# Posterior basolateral amygdala to ventral hippocampal CA1 drives approach behaviour to exert an anxiolytic effect

Guilin Pi [1,7], Di Gao [1,7], Dongqin Wu [1], Yali Wang[1,2], Huiyang Lei [1], Wenbo Zeng [3], Yang Gao [1], Huiling Yu [1], Rui Xiong [1], Tao Jiang[1], Shihong Li [1], Xin Wang [1], Jing Guo[1], Si Zhang [1], Taoyuan Yin [1], Ting He [1], Dan Ke[1], Ruining Li [1], Honglian Li[1], Gongping Liu[1], Xifei Yang [4], Min-Hua Luo [3], Xiaohui Zhang[5], Ying Yang [1]* & Jian-Zhi Wang [1,6]*

The basolateral amygdala (BLA) and ventral hippocampal CA1 (vCA1) are cellularly and functionally diverse along their anterior–posterior and superficial-deep axes. Here, we find that anterior BLA (aBLA) and posterior BLA (pBLA) innervate deep-layer calbindin1-negative (Calb1−) and superficial-layer calbindin1-positive neurons (Calb1+) in vCA1, respectively. Photostimulation of pBLA–vCA1 inputs has an anxiolytic effect in mice, promoting approach behaviours during conflict exploratory tasks. By contrast, stimulating aBLA–vCA1 inputs induces anxiety-like behaviour resulting in fewer approaches. During conflict stages of the elevated plus maze task vCA1$^{Calb1+}$ neurons are preferentially activated at the open-to-closed arm transition, and photostimulation of vCA1$^{Calb1+}$ neurons at decision-making zones promotes approach with fewer retreats. In the APP/PS1 mouse model of Alzheimer's disease, which shows anxiety-like behaviour, photostimulating the pBLA–vCA1$^{Calb1+}$ circuit ameliorates the anxiety in a Calb1-dependent manner. These findings suggest the pBLA–vCA1$^{Calb1+}$ circuit from heterogeneous BLA–vCA1 connections drives approach behaviour to reduce anxiety-like behaviour.

[1] Department of Pathophysiology, School of Basic Medicine, Key Laboratory of Ministry of Education of China and Hubei Province for Neurological Disorders, Tongji Medical College, Huazhong University of Science and Technology, Wuhan 430030, China. [2] Department of Physiology and Neurology, Key Laboratory for Brain Research of Henan Province, Xinxiang Medical University, Xinxiang 453000, China. [3] State Key Laboratory of Virology, CAS Center for Excellence in Brain Science and Intelligence Technology (CEBSIT), Wuhan Institute of Virology, Chinese Academy of Sciences, Wuhan 430071, China. [4] Key Laboratory of Modern Toxicology of Shenzhen, Shenzhen Centre for Disease Control and Prevention, 8 Longyuan Road, Nanshan District, Shenzhen 518055, China. [5] State Key Laboratory of Cognitive Neuroscience & Learning and IDG/McGovern Institute for Brain Research, Beijing Normal University, Beijing 100000, China. [6] Co-innovation Center of Neuroregeneration, Nantong University, Nantong 226001, China. [7] These authors contributed equally: Guilin Pi, Di Gao *email: yingyang@hust.edu.cn; wangjz@mail.hust.edu.cn

Emotion and decision-making are integral aspects of daily life that affect each other to help individuals adapt to stressful environments and can shape their life course. However, when individuals suffer from anxiety disorder, the most common maladaptive emotion, they generally prefer excessive avoidance decisions that can result in a pathological downward spiral[1]. The more often individuals avoid a situation, the less opportunity they have to learn that the real situation may be not as horrible as they feared[2,3]; furthermore, decreased approach exploration fails to endow them with the capacity to extinguish the previous fear, which can promote further anxiety[4,5]. Unfortunately, the mechanisms and the neural circuits by which avoidance-approach behaviours are controlled remain elusive, and this information could help break the abovementioned vicious cycle.

The basolateral amygdala (BLA) and hippocampus play essential roles in processing anxiety-associated events[6–10]. The non-specific activation of glutamatergic somata in the BLA, especially in its anterior part (aBLA), has an anxiogenic effect[4]. The selective stimulation of the BLA terminals in the central nucleus of the amygdala (CeA)[4] or the anterodorsal part of the bed nucleus of stria terminalis (adBNST)[11] instead has an anxiolytic effect. The ventral hippocampus (vHPC) is also closely related to anxiety[12]. Pharmacological lesions of the vHPC reduce anxiety-like behaviour[10]. When exposed to anxiogenic environments, theta-frequency firing within the vHPC is synchronised with medial prefrontal cortex (mPFC) discharge[13], and this functional connection preferentially represents anxiety-related behaviours[14,15]. By in vivo $Ca^{2+}$ imaging, cells involved in anxiety were discovered in ventral hippocampal CA1 (vCA1)[16]. The optogenetic inhibition of vCA1 anxiety cells reduces avoidance behaviour to produce anxiolytic effects[16]. The aBLA and vCA1 can orchestrate to modulate anxious states, and monosynaptic excitatory aBLA–vCA1 inputs exert anxiogenic effects by increasing avoidance behaviours[17]. Although projections from the posterior part of BLA (pBLA) to the vCA1 have been recently identified[18], whether pBLA targets vCA1 in a uniform or distinct pattern, as aBLA does in the aBLA–vCA1 connection, is largely unknown. Considering the functional diversity along the anterior–posterior axis of the BLA[19] and the superficial-deep axis of vCA1[16], it will be valuable to explore whether and how pBLA–vCA1 connections control approaching or avoiding under conflict situations to modulate anxiety-related behaviour. Furthermore, anxiety has been detected in early stages of Alzheimer's disease (AD) and can accelerate memory deficits[20], but the mechanisms underlying anxiety are not clear, and there is no efficient intervention.

In the present study, we dissected the structural and functional heterogeneities of BLA–vCA1 inputs. We found that pBLA- and aBLA-innervated calbindin1-positive neurons (Calb1+) in the superficial layer and calbindin1-negative neurons (Calb1−) in the deep layer of vCA1, respectively. Functionally, aBLA–vCA1 inputs were anxiogenic, while the pBLA–vCA1 inputs were anxiolytic, and stimulating vCA1$^{Calb1+}$ neurons could mimic the anxiolytic effect of pBLA–vCA1 inputs in an immediate, yet reversible, manner. Furthermore, the photostimulation of vCA1$^{Calb1+}$ neurons at the transition zones promoted approach behaviour with decreased retreat in approach-avoidance conflict paradigms, which partially explained the mechanisms underlying the anxiolytic effect of pBLA–vCA1 inputs. In APP/PS1 mice, a widely applied mouse model of AD, distinct changes in the protein network were found in aBLA and pBLA in response to amyloid-beta (Aβ) pathologies. Consistent with their differential proteomics, insufficient pBLA–vCA1 inputs and overactivated aBLA–vCA1 inputs with disorganized firing patterns were found in APP/PS1 mice. The activation of pBLA–vCA1 inputs remarkably increased approach behaviours and ameliorated anxiety and spatial memory deficits, while the simultaneous knockdown of Calb1 in vCA1 attenuated the beneficial effects of pBLA–vCA1 inputs both in wild-type and APP/PS1 mice, indicating the molecular mechanism of the pBLA–vCA1$^{Calb1+}$ circuit.

## Results

**a/pBLA innervates vCA1 along its superficial to deep axis.** To map the innervation between aBLA or pBLA and vCA1, we first used trans-synaptic virus-delivered anterograde trackers[21] (GFP-H129-G4 and mCherry-H129-R4) to outline the circuits in physiological conditions. After injecting GFP-H129-G4 into aBLA and pBLA, respectively (Supplementary Fig. 1a, f, i), we observed robust GFP expression in vCA1 but not in dorsal and intermediate hippocampal CA1 (Supplementary Fig. 1a–g). Interestingly, the innervation pattern was very different between aBLA and pBLA, i.e., aBLA predominantly innervated the deep layer (close to stratum oriens) (Supplementary Fig. 1d, k), while the pBLA innervated the superficial layer (close to stratum radiatum) (Supplementary Fig. 1h, l) of vCA1 pyramidal cells (PCs). This innervation pattern was substantially recapitulated when GFP-H129-G4 (green) and mCherry-H129-R4 (red) were respectively infused into aBLA and pBLA in the ipsilateral hemisphere of the same mouse (Supplementary Fig. 2a–c).

To further verify the intrinsic heterogeneity of direct monosynaptic inputs from aBLA and pBLA to vCA1, we employed anterograde monosynaptic transneuronal tracers (H129-ΔTK-tdT). The helper viruses (AAV-EF1a-DIO-TK-GFP and AAV-CaMKIIa-EGFP-P2A-Cre) were respectively injected into aBLA and pBLA to control initial herpes simplex virus (HSV) infection. Then, H129-ΔTK-tdT was injected stereotaxically into the same site. The helper virus allows the HSV spread anterogradely by one synapse. By detecting tdTomato in vCA1, we found that an ~91% projection signal from pBLA was detected in the superficial layer, and an ~89% projection signal from aBLA was observed in the deep layer of vCA1 (Fig. 1a, b). Compared with aBLA-innervated vCA1 neurons (aBLA–vCA1), the pBLA-innervated vCA1 neurons (pBLA–vCA1) were smaller and had more branched apical dendrites (Supplementary Fig. 3a, b). Approximately 41% of pBLA–vCA1 neurons possessed twin apical dendrites, and ~88% of aBLA–vCA1 neurons only had a single dendrite (Fig. 1c, d). These data outline the non-uniform structure of the BLA–vCA1 circuit along the anterior–posterior axis of BLA and the superficial-deep axis of vCA1.

Calbindin1 is one of the neuron markers in superficial layer of the hippocampus[22]. To further explore whether vCA1$^{Calb1+}$ neurons are the target of pBLA projections, we crossed Calb1-IRES2-Cre-D knock-in mice with the tdTomato reporter Ai9 to fluorescently label Calb1+ neurons (Supplementary Fig. 4) and anterogradely traced aBLA–vCA1 and pBLA–vCA1 circuits in Calb1-IRES2-Cre-D::Ai9 mice (Fig. 1e, f). We found that ~84% of pBLA–vCA1 neurons were Calb1+, while ~95% of aBLA–vCA1 neurons were Calb1− (Fig. 1g). The nature of pBLA-innervated neurons was further confirmed by co-staining with calbindin1 antibody in vCA1 after anterograde monosynaptic tracing (Supplementary Fig. 5). These data indicate that aBLA sends abundant innervations to vCA1$^{Calb1−}$ neurons, while the pBLA preferentially dominates vCA1$^{Calb1+}$ neurons.

Next, we detected the monosynaptic connections of the pBLA–vCA1$^{Calb1+}$ circuit by ex vivo brain slice recording (Fig. 1h–j, Supplementary Fig. 6). Upon the stimulation of pBLA–vCA1 inputs in the vCA1, robust responses were recorded from Calb1+, not Calb1−, neurons in the superficial layer of vCA1 pyramidal neurons (Fig. 1h, Supplementary Fig. 6b, d). Tetrodotoxin (TTX, 1 μM) and 4-amynopyridine (4AP, 100 μM)

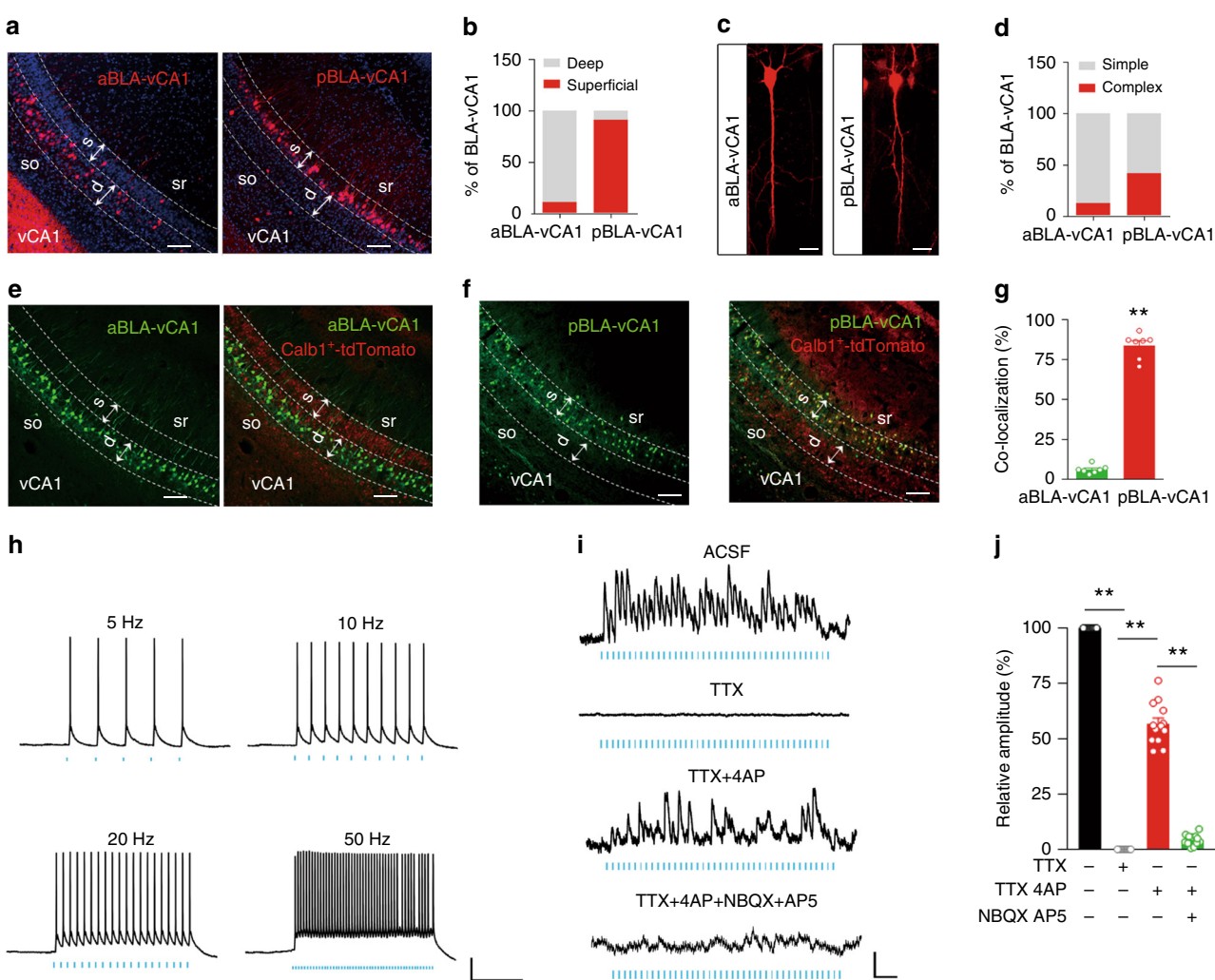

**Fig. 1 Features of a/pBLA-innervated vCA1 neurons along the superficial-deep axis.** Representative images (**a**) and quantification (**b**) show the predominance of aBLA-innervated neurons in the deep layer of vCA1 (left) and pBLA-innervated neurons in the superficial layer (right). By anterograde monosynaptic tracing, vCA1 neurons receiving monosynaptic inputs from aBLA and pBLA were labelled by tdTomato. aBLA anterior BLA, pBLA posterior BLA. Scale bar, 100 μm; $n = 4$ mice per group. Representative images (**c**) and quantification (**d**) show simple (**c**, left) and complex (**c**, right) vCA1 neurons innervated by aBLA and pBLA. Scale bar, 30 μm; $n = 5$ mice per group. **e–g** Distinct distributions of calbindin1-positive neurons (Calb1+) in aBLA- and pBLA-innervated vCA1 neurons. By anterograde multisynaptic tracing (H129-G4), pBLA-innervated vCA1 neurons (GFP) were predominately colocalized with Calb1+ (tdTomato) (**f**, **g**), while low colocalization was shown in aBLA-innervated vCA1 neurons (**e**, **g**). Scale bar, 100 μm; $n = 7$ mice per group. Unpaired $t$ test, $t = 24.76$ $df = 7.266$, $P < 0.0001$. **h** Action potential firing of a pBLA neuron in response to patterned blue laser light (472 nm, 5/10/20/50 Hz, 5 ms pulses) recorded by ex vivo current-clamp recording. Scale bar, 20 mv and 500 ms. **i, j** Subthreshold responses of vCA1 neurons to the photoactivation of pBLA–vCA1 inputs, 20 Hz, 5 ms pulses in ACSF (top) with TTX, TTX + 4AP or TTX + 4AP + AP5 + NBQX (GluR antagonist). Scale bar, 2 mv and 200 ms (**i**). Amplitude changes in EPSPs after TTX, TTX + 4AP or GluR antagonist perfusion (one-way ANOVA, $F_{(3, 48)} = 155.1$, $P < 0.0001$, Tukey's post hoc analysis, $P < 0.01$) (**j**). $n = 13$ cells from seven mice. Data are presented as the mean ± SEM. Source data are provided as a Source Data file.

were employed in the bath to remove any network activity. In pBLA-vCA1-ChR2 mice, light-induced EPSP, and EPSC amplitudes in vCA1$^{Calb1+}$ neurons persisted after TTX + 4AP perfusion, indicating direct, monosynaptic excitatory inputs from pBLA axon terminals (Fig. 1i, j, Supplementary Fig. 6a–c). Furthermore, the simultaneous inhibition of glutamate receptors by NBQX (2,3-Dioxo-6-nitro-1,2,3,4-tetrahydrobenzo [f] quinoxaline-7-sulphonamide; 20 μM) and AP5 (D-2-amino-5-phosphonopentanoate; 50 μM) abolished the light-induced excitation of the vCA1$^{Calb1+}$ neurons (Fig. 1i, j). Unlike pBLA, aBLA predominantly innervated the Calb1$^-$ neurons in the deep layer of vCA1 and established excitatory monosynaptic connection (Supplementary Fig. 6f–h). These data, together, suggest a monosynaptic excitatory connection from pBLA to vCA1 Calb1$^+$ neurons and aBLA to vCA1 Calb1$^-$ neurons.

**a/pBLA–vCA1 inputs exert heterogeneity in anxiety behaviour.** Then, we studied whether and how the different innervation patterns of pBLA–vCA1 and aBLA–vCA1 circuits result in functional heterogeneity in anxiety. First, we infused AAV5-CaMKIIa-eNpHR3.0-EYFP or AAV5-CaMKIIa-hChR2(H134R)-EYFP into the pBLA and bilaterally implanted optical fibres in vCA1 to target pBLA–vCA1 terminals by delivering light (Fig. 2a, g, Supplementary Fig. 7). Approach and/or avoidance behaviours were recorded in the elevated plus maze (EPM) and open field test (OFT), the recognised conflict exploratory tasks used to evaluate anxiety[23]. During the light-on epoch, we unexpectedly observed that the mice in the pBLA-vCA1-NpHR group, compared with eYFP control mice, explored open arms in EPM much less often (Fig. 2b, c) and seldom spent time in the centre during the OFT (Fig. 2d), indicating increased anxiety-related behaviours

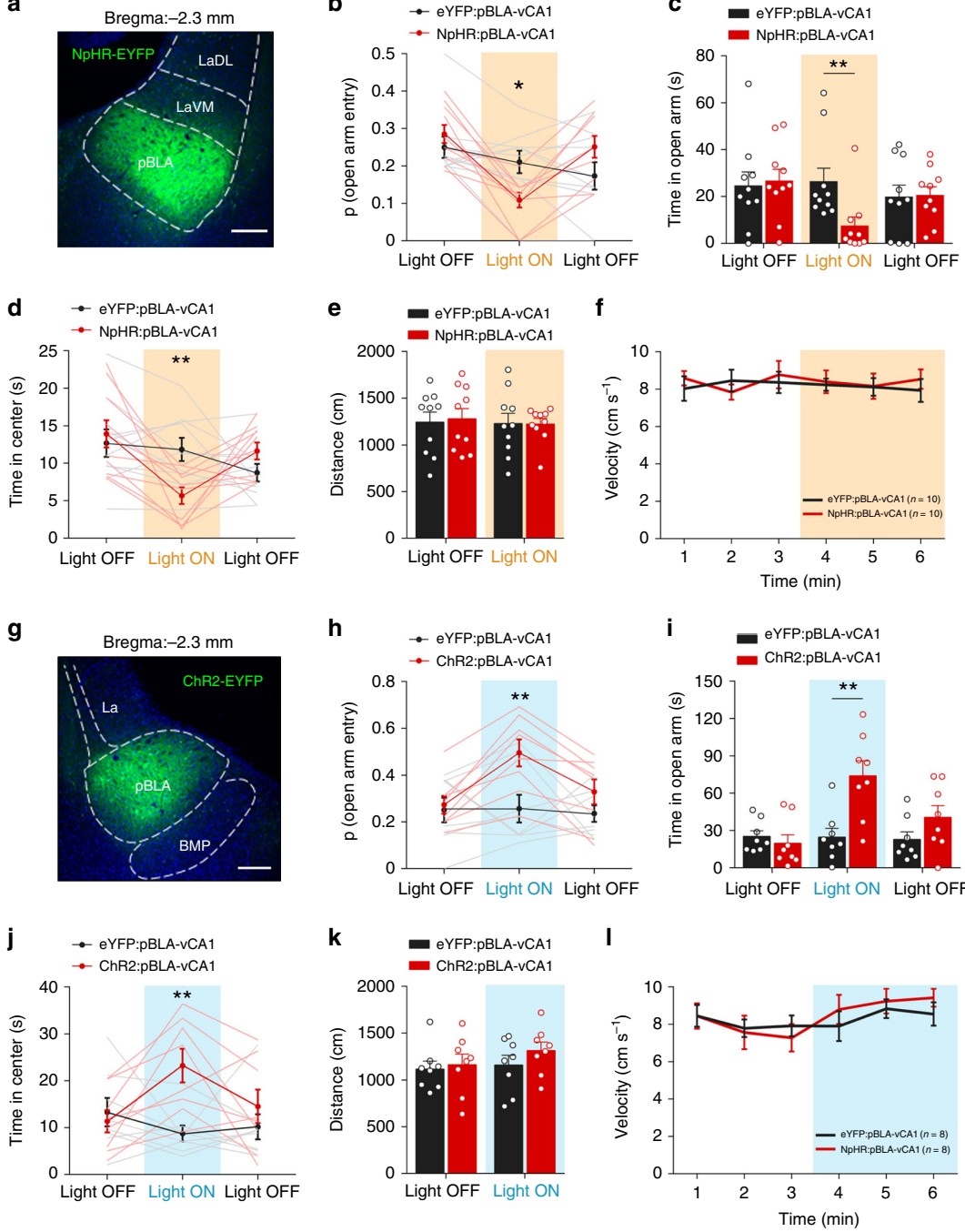

**Fig. 2 pBLA–vCA1 inputs promote approach behaviour and exert anxiolytic effects.** Representative confocal images confirmed the expression of NpHR (**a**) and ChR2 (**g**) in the pBLA. Scale bar, 200 μm. NpHR (**b–d**) and ChR2 (**h–j**) mice were tested in a single 9 min session in the EPM (**b, c, h, i**) or the OFT (**d, j**) with three 3-min epochs. **b, c** During the illumination epoch in the EPM, pBLA–vCA1 NpHR-eYFP optogenetic terminal stimulation decreased the probability of open-arm entry and time spent in open arms (two-way ANOVA group × epoch interaction; probability of open-arm entry $F_{(2, 36)} = 8.804$, $P = 0.0008$, Bonferroni post hoc analysis, *$P < 0.05$; time in open arms $F_{(2, 36)} = 5.115$, $P = 0.0111$, Bonferroni post hoc analysis, *$P < 0.05$). **d** Decreased time spent in the centre in the OFT during the NpHR illumination epoch (two-way ANOVA group × epoch interaction, $F_{(2, 36)} = 7.327$, $P = 0.0021$, Bonferroni post hoc analysis, *$P < 0.05$). **h, i** The increased probability of open-arm entry and time spent in the open arm detected during pBLA–vCA1 ChR2–eYFP optogenetic terminal stimulation (probability of open-arm entry $F_{(2, 28)} = 3.919$, $P = 0.0316$, Bonferroni post hoc analysis, **$P < 0.01$; time spent in open arms $F_{(2, 28)} = 6.299$, $P = 0.0055$, Bonferroni post hoc analysis, **$P < 0.01$). **j** Increased amount of time spent in the centre in the OFT during the ChR2 illumination epoch (two-way ANOVA group × epoch interaction, $F_{(2, 28)} = 6.844$, $P = 0.0038$, Bonferroni post hoc analysis, **$P < 0.01$). No difference in the distance travelled (**e, k**) or the velocity (**f, l**) was detected. Two-way ANOVA group × epoch interaction, distance travelled [NpHR] $F_{(1, 18)} = 0.06014$, $P = 0.8091$; [ChR2] $F_{(1, 14)} = 0.4512$, $P = 0.5127$; velocity [NpHR] $F_{(5, 90)} = 0.312$, $P = 0.9046$; [ChR2] $F_{(5, 70)} = 0.6561$, $P = 0.6579$. $n = 10$ mice (**b–f**) or 8 mice (**h–l**) per group. Data are presented as the mean ± SEM. Source data are provided as a Source Data file.

when inhibiting pBLA–vCA1 inputs. However, light stimulation in the pBLA-vCA1-ChR2 mice dramatically increased open-arm exploration, probability of open-arm entry and centre exploration (Fig. 2h–j), suggesting reduced anxiety when stimulating pBLA–vCA1 inputs. No locomotor changes were detected during the manipulation of pBLA–vCA1 inputs in the EPM and OFT paradigms (Fig. 2e, f, k, l). These data unexpectedly demonstrate an anxiolytic effect of pBLA–vCA1 excitation.

Considering that photostimulation can induce back-propagating action potentials that could contaminate the phenotypes of the circuit per se[24], we combined in vivo optogenetic manipulations with in vivo pharmacological manipulations to ensure the specificity of pBLA onto vCA1 inputs in the light-induced anxiolytic effect. ChR2 was specifically expressed in pBLA, and a guide cannula was accurately implanted in the vCA1. The glutamate receptor antagonists NBQX and AP5, or normal saline (Ctrl) were delivered through the cannula into the vCA1 30 min before illumination. The Ctrl mice phenocopied the light-increased approach behaviours during the EPM and OFT, while the administration of glutamate antagonists abolished the anxiolytic effect (Supplementary Fig. 8a–d). These data confirm that glutamatergic pBLA inputs onto vCA1 are sufficient to promote approach behaviour and to exert anxiolytic effects.

We then targeted aBLA–vCA1 inputs to explore their role in anxiety-associated behaviours (Fig. 3, Supplementary Fig. 9). The results showed that stimulation during the light-on epoch (472 nm) remarkably decreased open-arm and centre exploration, with fewer approaches or increased avoidance during the EPM and OPT, respectively (Fig. 3a–d), suggesting that the activation of aBLA–vCA1 inputs results in an anxiogenic effect. However, the inhibition of aBLA–vCA1 inputs (589 nm) remarkably increased open-arm and centre exploration during the EPM and OPT (Fig. 3g–j), suggesting an anxiolytic effect. No difference in locomotion was detected across the groups (Fig. 3e, f, k, l).

Together, these data demonstrate an opposite role of pBLA–vCA1 and aBLA–vCA1 inputs in modulating anxiety-like behaviours, i.e., pBLA–vCA1 inputs are anxiolytic, and aBLA–vCA1 inputs are anxiogenic.

**vCA1$^{Calb1+}$ neurons control approach-avoidance behaviours.** To explore the mechanisms underlying the anxiolytic effect of pBLA–vCA1 stimulation during decision-making in conflict situations, we analysed the firing activity of vCA1$^{Calb1+}$ neurons during conflict exploratory tasks in freely moving mice (Fig. 4a). The GCaMP6f or the empty GFP-vector were expressed in vCA1$^{Calb1+}$ neurons using a Cre-dependent strategy (Supplementary Fig. 10), and the population fluorescence in three defined periods was measured: baseline (a 5 s period beginning 15 s before the mouse entered the closed or open arms), pre-entry (a 5 s period beginning 8 s before the mouse entered the closed or open arms) and after-entry (a 1 s period after the mouse entered closed or open arms) (Fig. 4b, e). On average, calcium activity increased when the mice moved from an open arm into a closed arm relative to the baseline (Fig. 4b–d), while calcium activity decreased when the mice moved from a closed arm to an open arm (Fig. 4e–g). No significant fluctuation in calcium activity was detected in vCA1$^{Calb1+}$-GFP control neurons (Fig. 4d, g, and Supplementary Fig. 11a, b). Furthermore, the average calcium activity (5 s period around the transition point) in the closed arm was much higher than that in the open arm (Fig. 4h). These data suggest that the activation of vCA1$^{Calb1+}$ neurons innervated preferentially by pBLA is involved in avoidance or approach decision-making based on the assessment of aversion/safe information in conflict situations.

To verify whether the optogenetic manipulation of vCA1$^{Calb1+}$ neurons can shift avoidance-approach balance and, thus, modulate anxiety-related behaviours, ChR2–eYFP fluorescent fusion protein was targeted to vCA1$^{Calb1+}$ neurons using a Cre-dependent strategy in Calb1-IRES2-Cre-D knock-in mice (Supplementary Fig. 12). The photoactivation of vCA1$^{Calb1+}$ neurons robustly increased approach behaviours, evidenced by an increase in the centre exploration time in the OFT and an increase in open-arm exploration time and open-arm entry probability in the EPM test (Supplementary Fig. 12a–c). The light effects of vCA1$^{Calb1+}$ neurons were not due to changes in locomotor activity, as the moving distance did not change (Supplementary Fig. 12d). These data suggest that vCA1$^{Calb1+}$ neurons can promote approach behaviours in conflict exploratory tasks to exert an anxiolytic effect following pBLA–vCA1 input activation.

To further investigate how vCA1$^{Calb1+}$ neurons drive approach behaviours, we established an L-type elevated maze that consists of one open arm and one closed arm (Fig. 4i, m). Compared with the traditional EPM, the L-type maze has a definite moving direction that can help investigators accurately determine the animals' decision-making behaviours. In the L-type maze, we specifically targeted vCA1$^{Calb1+}$ neurons by delivering a brief stimulation (2 s at 8 mW) to vCA1$^{Calb1+}$-ChR2 mice when they approached the boundary between the open and the closed arms (Fig. 4i–o). When moving from the closed arm to the open arm, photoactivation at the transition site significantly increased retreat latency (Fig. 4i, j), decreased approach latency to the open arm (Fig. 4k) and had no effect on the staying time in the closed arm (Fig. 4l). Then, we performed a similar stimulation protocol in these mice when they moved from the open arm towards the closed arm (Fig. 4m). In this case, photostimulation decreased staying time in the closed arm (Fig. 4n) and had no effect on latency to enter the closed arm (Fig. 4o). These data indicate that vCA1$^{Calb1+}$ neurons bias decision-making towards approach in conflict exploratory tasks.

These data together suggest that vCA1$^{Calb1+}$ neurons phenocopy the anxiolytic effect of pBLA–vCA1 inputs by controlling approach-avoidance behaviours.

**pBLA–vCA1$^{Calb1+}$ circuit ameliorates anxiety in APP/PS1 mice.** A previous study showed that Aβ accumulation in the BLA enhances anxiety-like behaviour in AD transgenic mice[25]. To explore whether and how aBLA or pBLA is involved in AD-related anxiety, we first examined Aβ pathologies in the aBLA and pBLA of APP/PS1 mice carrying mutated APP and PS1 genes, overproducing Aβ in the brain. Compared with the age- and sex-matched wild-type controls, APP/PS1 mice (6 months old) showed robust intraneuronal Aβ and extra-neuronal amyloid plaques in pBLA and aBLA detected by 6E10 staining (Fig. 5a). To gain further insight into the difference between aBLA and pBLA in mice, we performed an unbiased proteomic analysis (Fig. 5b–f, Supplementary Fig. 13). In wild-type mice, 5079 and 5083 unique proteins were identified in the aBLA and pBLA, respectively (Supplementary Fig. 13a). In addition to 5072 common proteins, the aBLA had seven distinct proteins (Table 1), i.e., deoxyuridine triphosphatase, serine/threonine-protein phosphatase 1 regulatory subunit 10, RAB6-interacting golgin, acyl-coenzyme A thioesterase 8, haemoglobin subunit beta-2, uncharacterised protein KIAA1467, and reticulocalbin-3, while the pBLA had 11 characteristic proteins (Table 2), i.e., E3 ubiquitin-protein ligase RNF181, cAMP and cAMP-inhibited cGMP 3′,5′-cyclic phosphodiesterase 10 A, carboxypeptidase N subunit 2, pre-mRNA 3 end processing protein WDR33, conserved oligomeric Golgi complex subunit 5, transcription factor Sp8, Eukaryotic translation initiation factor 4E-binding protein 2,

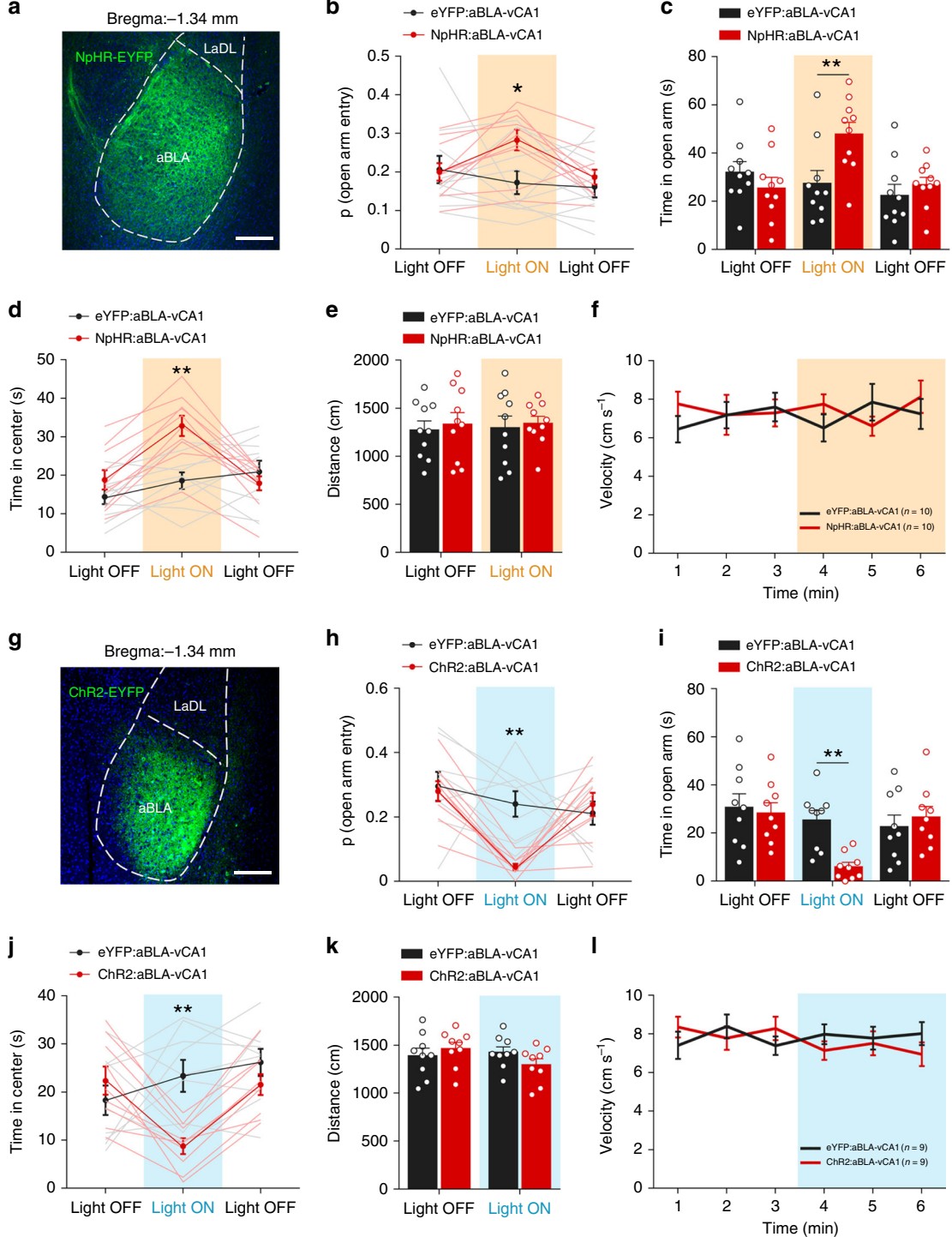

**Fig. 3 aBLA–vCA1 inputs promote avoidance behaviour and exert anxiogenic effects. a, g** Representative confocal images of ChR2 (**a**) and NpHR (**g**) expression in the aBLA. Scale bar, 200 μm. ChR2 (**b–d**) and NpHR (**h–j**) mice were tested in the EPM (**b, c, h, i**) and OFT (**d, j**) as described in Fig. 2. During the illumination epoch in the EPM, the probability of open-arm entry and time spent in the open arms decreased in ChR2 mice (**b, c**) but increased in NpHR mice (**h, i**) compared with eYFP mice (two-way ANOVA group × epoch interaction; probability of open-arm entry [ChR2] $F_{(2, 32)} = 5.477$, $P = 0.0090$, Bonferroni post hoc analysis, **$P < 0.01$; [NpHR] $F_{(2, 36)} = 3.409$, $P = 0.0441$, Bonferroni post hoc analysis, *$P < 0.05$; time spent in open arms [ChR2] $F_{(2, 32)} = 4.449$, $P = 0.0197$, Bonferroni post hoc analysis, **$P < 0.01$; [NpHR] $F_{(2, 36)} = 6.815$, $P = 0.0031$, Bonferroni post hoc analysis, **$P < 0.01$. During the illumination epoch in the OFT, the time spent in the centre decreased in ChR2 mice (**d**) but increased in NpHR mice (**j**) (two-way ANOVA group × epoch interaction, ChR2: $F_{(2, 32)} = 9.428$, $P = 0.0006$, Bonferroni post hoc analysis, **$P < 0.01$; NpHR: $F_{(2, 36)} = 14.27$, $P < 0.0001$, Bonferroni post hoc analysis, **$P < 0.01$). No effects of light stimulation on distance travelled (**e, k**) or velocity (**f, l**) were detected. Two-way ANOVA group × epoch interaction, travelled distance [ChR2] $F_{(1, 16)} = 2.312$, $P = 0.1479$; [NpHR] $F_{(1, 18)} = 0.007063$, $P = 0.9340$; velocity [ChR2] $F_{(5, 80)} = 1.13$, $P = 0.3514$; [NpHR] $F_{(5, 90)} = 1.188$, $P = 0.3212$. $n = 10$ mice (**b–f**) or 9 mice (**h–l**) per group. Data are presented as the mean ± SEM. Source data are provided as a Source Data file.

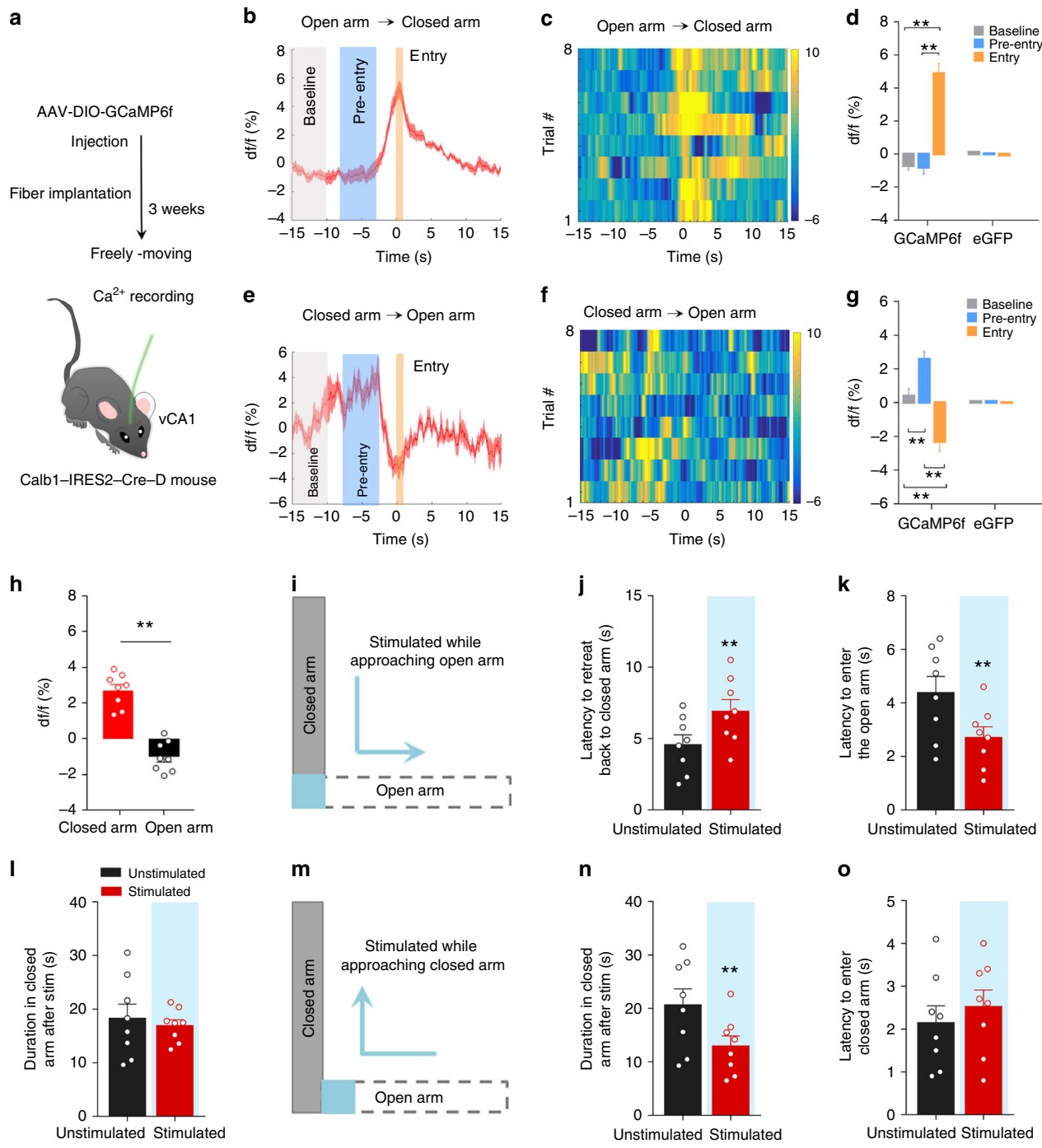

**Fig. 4 vCA1$^{Calb1+}$ neurons promote approach in conflict exploratory tasks. a** Schematic of Ca$^{2+}$ recording in the EPM. **b, e** Average Ca$^{2+}$ transients from vCA1$^{Calb1+}$ neurons during the behavioural transition between closed-arm and the open-arm compartments (transition point at time 0 s) ($n = 7$ or eight mice per group). **c, f** Heat maps of normalised Ca$^{2+}$ activity in vCA1$^{Calb1+}$ neurons during transition, as shown in **b** and **e**. Average df/f for baseline, pre-entry, and entry periods in GCaMP6f mice and control GFP mice during transition from open to closed arms (**d**) or closed to open arms (**g**). Two-way ANOVA, closed to open arm: $F_{(2, 26)} = 17.63$, $P < 0.0001$, Tukey's multiple comparisons test, **$P < 0.01$; open to closed arm: $F_{(2, 26)} = 47.83$, $P < 0.0001$, Tukey's multiple comparisons test, **$P < 0.0001$. **h** Average df/f of closed arm and open arm (5 s period around transition point) in GCaMP6f mice. Unpaired $t$ test, $t = 8.246$ $df = 14$, **$P < 0.0001$. Schematic of stimulation while moving from closed arms (**i**) or open arms (**m**) to the transition zone. **j** Latency retreated to the closed arm in stimulated and unstimulated trials. Paired $t$ test, $t = 4.225$ $df = 7$, **$P = 0.0039$. **k** Latency to enter the open arm following stimulation. Paired $t$ test, $t = 5.608$ $df = 7$, **$P = 0.0008$. **l** Duration in the closed arm for stimulated and unstimulated trials. Paired $t$ test, $t = 0.8115$ $df = 7$, $P = 0.4438$. **n** Same data format as **l**, but for stimulation while moving from the open arms to the transition zone. Paired $t$ test, $t = 5.445$ $df = 7$, **$P = 0.0010$. **o** Latency to enter the closed arm following stimulation. Paired $t$ test, $t = 1.661$ $df = 7$, $P = 0.1407$. $n = 8$ mice per group (**h–l**, **n**, **o**). Data are presented as the mean ± SEM. Source data are provided as a Source Data file.

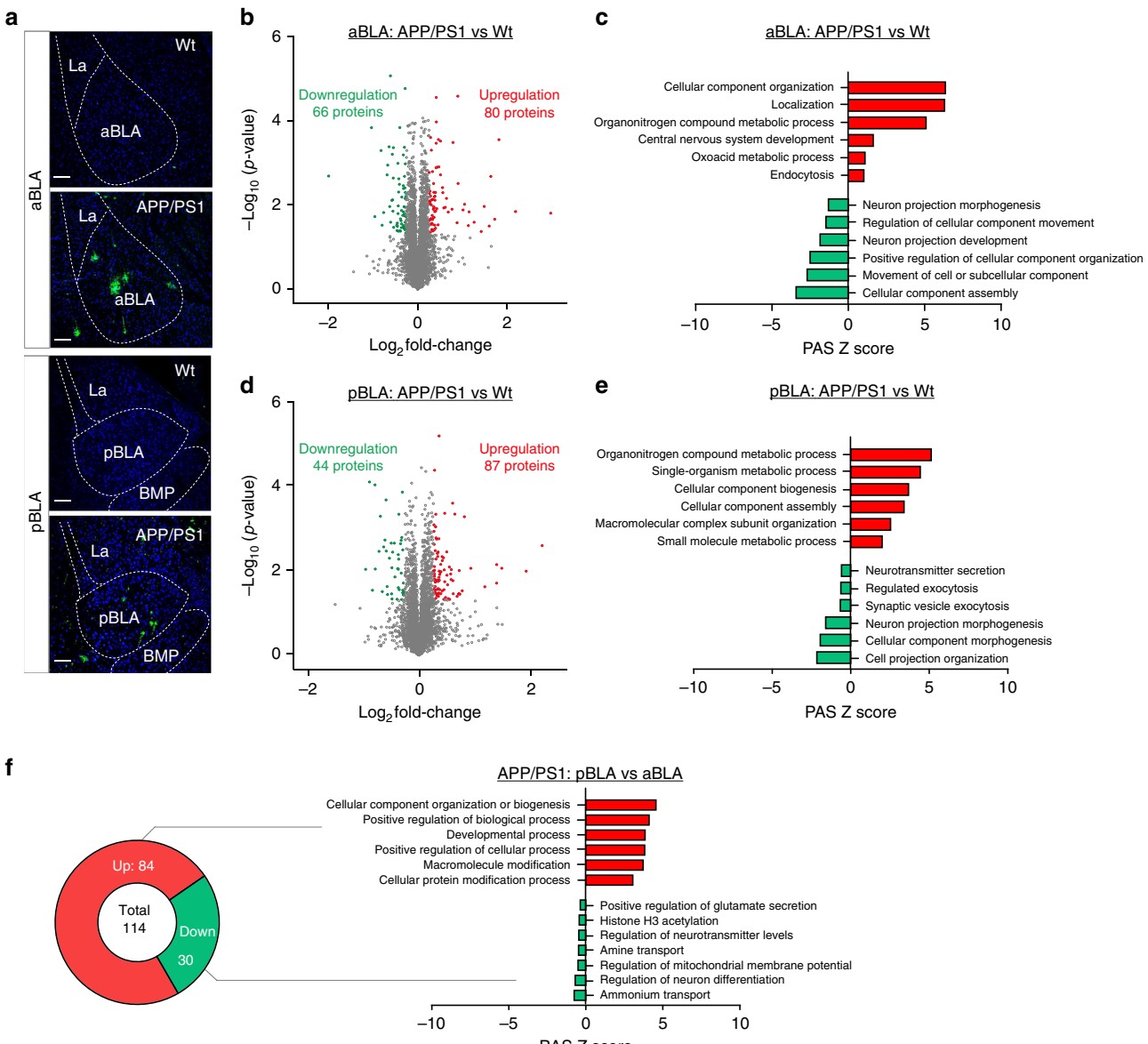

**Fig. 5 Proteomics reveals distinct molecular changes in the aBLA versus pBLA of APP/PS1 mice. a** Representative images of Aβ pathology in the aBLA and pBLA by 6E10 staining. Scale bar, 100 μm. Volcano plot showing protein changes in the aBLA (**b**) and pBLA (**d**) in 6-month-old APP/PS1 versus wild-type (Wt) mice. Red and green dots represent proteins whose abundance is significantly increased or decreased. GO enrichment analysis of up- and downregulated GO categories in the aBLA (**c**) and pBLA (**e**) of APP/PS1 mice versus Wt mice, or GO categories differentially regulated in pBLA versus aBLA of APP/PS1 mice (**f**). $n = 3$ mice per group.

**Table 1 Distinct proteins in aBLA.**

| Protein IDs | Protein names | Gene names |
|---|---|---|
| Q9JJ44;Q9CQ43;Q8VCG1 | Deoxyuridine triphosphatase | Dut |
| Q99KB0;Q3ULI5;Q80W00 | Serine/threonine-protein phosphatase 1 regulatory subunit 10 | Ppp1r10 |
| Q8BRM2 | RAB6-interacting golgin | Gorab |
| Q3U965;P58137;G3UXJ8;Q8BZR4 | Acyl-coenzyme A thioesterase 8 | Acot8 |
| D4N6U4;Q9QUN8;Q54AH9;Q549D9;D4N6N9;D4N6M2;D0U293; D0U285;D0U284;A8DV41;P02089;D4N6N2;A8DV59 | Haemoglobin subunit beta-2 | Hbbt1;Hbb-b2;Hbbt2 |
| B2RPZ7;Q8BYI8;Q99L10 | Uncharacterised protein KIAA1467 | 8430419L09Rik;Kiaa1467 |
| A0A1B0GS22;A0A1B0GSK5;A0A1B0GR19; A0A1B0GR86;Q8BH97 | Reticulocalbin-3 | Rcn3 |

**Table 2 Distinct proteins in pBLA.**

| Protein IDs | Protein names | Gene names |
|---|---|---|
| S4R2J5;D3YUJ1;Q9CY62 | E3 ubiquitin-protein ligase RNF181 | Rnf181 |
| S4R197;B2LYU5 | cAMP and cAMP-inhibited cGMP 3′,5′-cyclic phosphodiesterase 10A | Pde10a |
| Q9DBB9 | Carboxypeptidase N subunit 2 | Cpn2 |
| Q8VE87;Q8K4P0 | Pre-mRNA 3 end processing protein WDR33 | Wdr33 |
| Q8C0L8 | Conserved oligomeric Golgi complex subunit 5 | Cog5 |
| Q640M8;Q5QR90;Q8BMJ8 | Transcription factor Sp8 | Sp8 |
| Q3UZD4;Q3UFP6;P70445 | Eukaryotic translation initiation factor 4E-binding protein 2 | Eif4ebp2 |
| Q3TBV3;Q3URQ4 | Uncharacterised protein | C78339 |
| B2KFS7;Q8BFQ9 | Kelch-like protein 42 | Klhl42 |
| A2A8F6;Q9D6I9 | Leucine rich adaptor protein 1 | Lurap1 |
| A0A171EBL2;E9Q555 | E3 ubiquitin-protein ligase RNF213 | Rnf213 |

uncharacterised protein, Kelch-like protein 42, leucine rich adaptor protein 1, and E3 ubiquitin-protein ligase RNF213. Among their common proteins, 59 proteins were significantly increased and 27 proteins were decreased in the pBLA compared with levels in the aBLA (Supplementary Fig. 13b). Gene ontology (GO) enrichment analyses suggested that many of the upregulated biological processes in the pBLA were related to single-organism processes, biological regulation, metabolic processes, etc., while downregulated biological processes in the pBLA were associated with cell differentiation, positive regulation of transport, regulation of protein localisation, etc. (Supplementary Fig. 13c). These data further support the natural heterogeneity of the BLA along its anterior–posterior axis. Looking for changes in the aBLA and pBLA protein networks in AD, we compared APP/PS1 mice with wild-type controls and found that 80 proteins increased and 66 proteins decreased in the AD aBLA, while 87 increased and 44 declined in the AD pBLA, relative to levels in the wild-type mice (Fig. 5b, d). GO analyses suggested that single-organism processes, biological regulation, cellular component organisation and metabolic processes were the top items upregulated in both the aBLA and pBLA of APP/PS1 mice compared with those affected in wild-type mice (Fig. 5c, e). In contrast, single-organism developmental processes, positive regulation of cellular processes, cellular component biogenesis, etc. were downregulated in the aBLA, while anatomical structure morphogenesis, cell development, response to chemicals, etc. were downregulated in the pBLA (Fig. 5c, e). Interestingly, comparing the pBLA and aBLA revealed that cation transport and cell activation functions were relatively higher in the pBLA of wild-type mice (Supplementary Fig. 13c), but these functions were changed in the opposite direction in APP/PS1 mice (Fig. 5f). These data, together, indicate that the difference in the protein network between aBLA and pBLA may result in their distinct responses to Aβ pathology in AD.

Then, we asked how aBLA–vCA1 and pBLA–vCA1 inputs changed in AD and whether they were involved in AD anxiety. GCaMP6f were precisely expressed in aBLA and pBLA neurons, which were shown to innervate vCA1 neurons by injecting rAAV-hSyn-Cre into vCA1 and AAV-EF1a-DIO-GCaMP6f into aBLA and pBLA, respectively. In the EPM test, wild-type mice exhibited a 5.3% increase in calcium activity in the aBLA–vCA1 circuit during movement into the open arms (Fig. 6a). In contrast, APP/PS1 mice exhibited a distinct pattern in which increased calcium signals skewed towards the periods of pre-entry and showed a scattered distribution in the period after entering into the open arm (Fig. 6a, b). These data suggest that the aBLA–vCA1 circuit is abnormally activated in AD under approach-avoidance conflict. For population calcium recording in the pBLA–vCA1 circuit at its entry node, we observed a significant increase in pre-entry and a decline in the entry period as wild-type mice moving

from a closed arm to an open arm (Fig. 6c, d), which was consistent with the observations in vCA1$^{Calb1+}$ neurons at the exit node of the pBLA–vCA1 circuit (Fig. 4e–g). However, the high calcium signals locked in the pre-entry period were significantly diminished in APP/PS1 mice (Fig. 6d). These data suggest disorganised firing in both aBLA–vCA1 and pBLA–vCA1 circuits under approach-avoidance conflict in AD. To verify whether the locomotor activity was involved in the disorganised firing seen in APP/PS1 mice, we analysed the moving velocity. The velocities at baseline and in the pre-entry period were much slower than that at the entry period in wild-type and APP/PS1 mice (Supplementary Fig. 14a, c, d, f), indicating risk-assessment behaviour and decision-making hesitation between approach and avoidance in conflict situation. However, no difference in average moving speed was detected between wild-type and AD mice (Supplementary Fig. 14c, f), which confirmed that the disorganised firing was not caused by locomotor variables.

Then, we used retrograde tracing (CTB, Cholera toxin subunit B) and c-Fos co-staining to evaluate the overall alterations in aBLA–vCA1 and pBLA–vCA1 circuits in APP/PS1 mice during the EPM test. In wild-type mice, both c-Fos$^+$ and CTB$^+$ neurons were much more abundant in pBLA than in aBLA (Fig. 6e), indicating a stronger activation of the pBLA–vCA1 circuit in normal conditions, which was consistent with the upregulation of cell activation in pBLA in the proteomic analysis (Supplementary Fig. 13c). In line with the downregulation of cation transport, cell activation and the positive regulation of glutamate secretion in the pBLA of APP/PS1 mice (Fig. 5f), both c-Fos$^+$ and CTB$^+$ neurons in the pBLA were barely detectable in APP/PS1 mice (Fig. 6e), suggesting that an insufficient activation of the pBLA–vCA1 circuit can skew the balance between aBLA–vCA1 and pBLA–vCA1. After the photostimulation of the pBLA–vCA1 circuit (Fig. 6f), inhibition in APP/PS1 mice was rescued, as evidenced by the increased activation of Calb1$^+$ neurons (Supplementary Fig. 15a, b) in the vCA1 without changes in Calb1 expression (Supplementary Fig. 15c, d). Furthermore, the photoactivation of pBLA–vCA1 inputs in APP/PS1 mice increased open-arm entry probability and open-arm exploration time in the EPM test (Fig. 6i, j) with an increased centre exploration time in the OFT (Fig. 6k). These data suggest that activating pBLA–vCA1 inputs could ameliorate anxiety in AD mice.

Expressing Calb1 is the main difference between aBLA–vCA1 and pBLA–vCA1 circuit output nodes. To explore the role of Calb1 in the anxiolytic effect of the pBLA–vCA1 circuit, we microinjected adeno-associated virus (AAV)-shCalb1 or the control AAV-shNT (encoding a nontargeting shRNA) into the vCA1 of wild-type mice and measured anxiety behaviours (Supplementary Fig. 16a–c). The reduction in Calb1 levels (~40%) was confirmed by western blotting (Supplementary

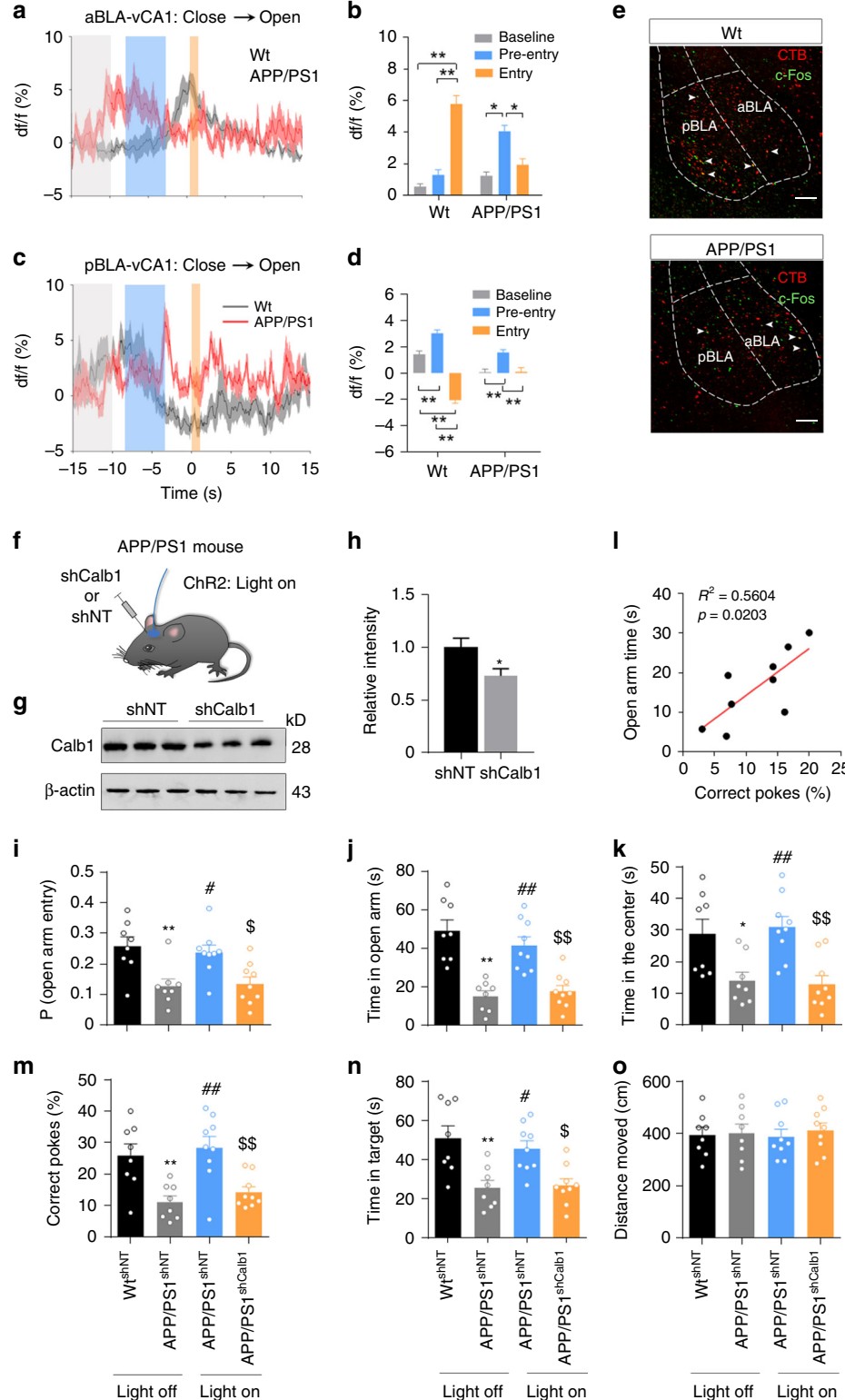

Fig. 14b, c). Simultaneously, the anxiolytic effect induced by pBLA–vCA photostimulation was abolished, as evidenced by the reduced open-arm entries and open-arm exploration time in EPM (Supplementary Fig. 16d, e) and the reduced centre exploration time in OFT (Supplementary Fig. 16f). No difference in motor function was detected among the groups (Supplementary Fig. 16g). Furthermore, downregulating Calb1 by the injection of AAV-shCalb1 in the vCA1 (Fig. 6f–h) abolished

the anxiolytic effects of pBLA–vCA photostimulation in APP/PS1 mice as measured by the EPM (Fig. 6i, j) and OFT (Fig. 6k). These data, together, indicate that Calb1 is required for the anxiolytic effect of pBLA–vCA1 circuit stimulation in both wild-type and AD mice.

Given that anxiety can exacerbate memory deficits in AD[20,26], we administered the Barnes maze (BM) to test spatial cognitive functions in APP/PS1 mice. Improved performance during

**Fig. 6 Activation of pBLA–vCA1$^{Calb1}$ circuit ameliorates anxiety in APP/PS1 mice.** Average Ca$^{2+}$ transients from aBLA–vCA1 (**a**) and pBLA–vCA1 (**c**) neurons in wild-type (Wt) or APP/PS1 mice during transition from closed to open arms (transition point at 0 s). **b, d** Average df/f for baseline, pre-entry, and entry periods in Wt$^{GCaMP6f}$ and APP/PS1$^{GCaMP6f}$ mice during transition. Two-way ANOVA; aBLA–vCA1: $n = 10$ mice per group, $F_{(2, 36)} = 47.57$, $P <$ 0.0001, Tukey's analysis, *$P < 0.05$, **$P < 0.01$; pBLA–vCA1: $n = 11$ mice per group, $F_{(2, 40)} = 34.18$, $P < 0.0001$, Tukey's analysis, **$P < 0.01$. **e** Representative co-staining of CTB+/c-Fos+ neurons in the aBLA and pBLA shown by vCA1 injection of CTB (red) and measured at 90 min after EPM. Scale bar, 100 μm. **f** Schematic of pBLA–vCA1 photostimulation and vCA1 injections in APP/PS1$^{ChR2}$ mice. **g, h** AAV-ShCalb1-induced reduction in Calb1 confirmed by western blotting. Unpaired $t$ test, $t = 4.136$, $df = 3.764$, $P = 0.0163$. $n = 3$ per group. **i, j** In the EPM, ChR2-APP/PS1-shCalb1 mice show fewer open-arm entries and less time spent in the open arm during the 5 min illumination epoch. One-way ANOVA, $F_{(3, 30)} = 7.106$, $P = 0.001$ (**i**) and $F_{(3, 30)} =$ 16.94, $P < 0.0001$ (**j**). **k** In the OFT, the photostimulation of pBLA–vCA1 inputs increased time spent in the centre in APP/PS1-shNT but not APP/PS1-shCalb1 mice (one-way ANOVA, $F_{(3, 30)} = 7.972$, $P = 0.0005$. **l** Positive correlation between time spent in the open arms in the EPM and correct pokes in the BM. **m–o** In the BM, photostimulation of pBLA–vCA1 inputs increased correct pokes (**m**) and target time (**n**) in APP/PS1-shNT but not APP/PS1-shCalb1 mice without changing distance travelled. One-way ANOVA, $F_{(3, 30)} = 8.729$, $P = 0.0003$ (correct pokes) or $F_{(3, 30)} = 8.231$, $P = 0.0004$ (time in target) or $F_{(3, 30)} = 0.1187$, $P = 0.9484$ (distance travelled). $n = 8$ or 9 mice per group, Tukey's multiple comparisons test, *$P < 0.05$, **$P < 0.01$ versus WT-shNT-light off; #$P < 0.05$, ##$P < 0.01$ versus APP/PS1-shNT-light off; $$P < 0.05$, $$$P < 0.01$ versus APP/PS1-shNT-light on (**i-k, m-o**). Data are presented as the mean ± SEM. Source data are provided as a Source Data file.

training days was identical in wild-type and APP/PS1 mice (Supplementary Fig. 17a, b). In the probe trial, the APP/PS1 mice showed less time in the target quadrant and fewer correct pokes than the wild-type mice (Supplementary Fig. 17c, d), indicating memory deficits. No motor dysfunction was detected, as evidenced by the identical distance moved by both groups in the BM (Supplementary Fig. 17e). Interestingly, anxiety status was significantly correlated with spatial memory deficits in 6–8 months APP/PS1 mice (Fig. 6l). In the probe trial of the BM, the photoactivation of pBLA–vCA1 inputs robustly increased time in the target and correct pokes without changing motor ability in APP/PS1 mice, and these improvements in AD memory deficits were significantly abolished by Calb1 downregulation in the vCA1 (Fig. 6m–o). These data demonstrate that deficits of the pBLA–vCA1$^{Calb1}$ circuit in AD mice links anxiety and memory impairment and that the activation of the pBLA–vCA1$^{Calb1}$ circuit can rescue anxiety-associated memory deficits in a Calb1-dependent manner.

## Discussion

Avoidance-approach imbalance is a core feature of anxiety[1,27], and regaining balance serves as a central psychological strategy for anxiety treatment[28]. The BLA and vCA1 can orchestrate anxiety via their direct connections[17], but whether and how BLA–vCA1 inputs control avoidance-approach behaviour is unknown. Recently, topographical features along the anterior–posterior axis of the BLA and superficial-deep axis of vCA1 have been disclosed[16,18,19]. However, whether BLA–vCA1 inputs are structurally and functionally uniform along the anterior–posterior axis of the BLA and the superficial-deep axis of vCA1 remain unclear. In the present study, we found that neurons in the aBLA and pBLA innervated the deep layer Calb1$^-$ neurons and the superficial layer Calb1$^+$ neurons of vCA1, respectively, revealing the structural heterogeneity of BLA–vCA1 circuits. Functionally, we demonstrated that the aBLA–vCA1 inputs were anxiogenic, while pBLA–vCA1 inputs were anxiolytic. In mechanism studies, we revealed that the optogenetic activation of pBLA–vCA1 inputs, or simply stimulating vCA1-Calb1$^+$ neurons, remarkably increased approach behaviour with decreased avoidance and retreat in conflict decision-making tests, while the inhibition of pBLA–vCA1 inputs robustly decreased approach behaviour (Fig. 7). To the best of our knowledge, this is the first report aiming to uncover the structural and functional heterogeneity of aBLA–vCA1 and pBLA–vCA1 circuits. We also found a highly differential protein network between the aBLA and pBLA, with disorganised firing of a/pBLA–vCA1 inputs in APP/PS1 mice, and the optogenetic stimulation of the

pBLA–vCA1$^{Calb1+}$ circuit attenuated anxiety behaviours and spatial memory deficits in a Calb1-dependent manner. These data indicate that input specificity and Calb1 levels are crucial in the pBLA–vCA1$^{Calb1+}$ circuit stimulation-induced amelioration of anxiety and anxiety-associated memory deficits in AD mice. Thus, the pBLA–vCA1$^{Calb1+}$ circuit may serve as a potential target for intervention in neuropsychiatric or neurodegenerative disorders, such as AD.

Manipulating aBLA and posterior BLA (pBLA or BLP) can elicit negative and positive emotional behaviours[19], respectively, but their heterogeneity at the molecular level is still unclear. Our proteomic data revealed that 86 proteins were differentially expressed within the aBLA and pBLA. In addition, we identified 7 aBLA- and 11 pBLA-characterised proteins. This novel differential protein outline in BLA subregions can help us better understand the distinct responses of aBLA and pBLA in neurological disorders.

Recent studies have shown that both the aBLA and pBLA send projections to vCA1[17,18]. Here, we found that aBLA–vCA1 and pBLA–vCA1 inputs were parallel and largely non-overlapping. In addition to the difference along the anterior–posterior axis of the BLA, aBLA- and pBLA-dominated vCA1 neurons were fairly distinct. Spatially, the aBLA neurons predominantly innervate Calb1$^-$ neurons in the deep layer of vCA1 PCs, while the pBLA neurons robustly projected to Calb1$^+$ neurons in the superficial layer of vCA1 PCs. Morphologically, vCA1$^{Calb1+}$ neurons innervated by the pBLA are much smaller and more complex than vCA1$^{Calb1-}$ neurons innervated by aBLA. This distribution pattern and the morphological characteristics of vCA1$^{Calb1+}$ neurons are consistent with the findings of the Calb1$^+$ neurons in dorsal CA1[22]. Thus, our current study reveals vCA1 heterogeneity, not only across the layers but also across inputs, suggesting differential information flow through these two non-overlapping circuits.

By dissecting the anterior and posterior parts of the BLA, we found that stimulating aBLA–vCA1 inputs was anxiogenic, while stimulating pBLA–vCA1 glutamic inputs was anxiolytic. Our data on aBLA–vCA1 inputs were consistent with those presented by Felix-Ortiz et al., in which virus was injected in the pars anterior of the BLA[17]. Including vCA1 neurons, other downstreams of a/pBLA, such as mPFC[29], BNST and CeA were also related with emotional behaviours[4,11]. Thus, the anxiolytic results from the inhibition of whole BLA should be carefully interpreted, such as to what extend aBLA and pBLA were inhibited, and how the balance between anxiogenic and anxiolytic downstream effectors of a/pBLA shifted after BLA inhibition. Using in vivo calcium recording[30], we observed that both the entry and exit nodes of the pBLA–vCA1 circuit, i.e., pBLA neurons and vCA1$^{Calb1+}$ neurons,

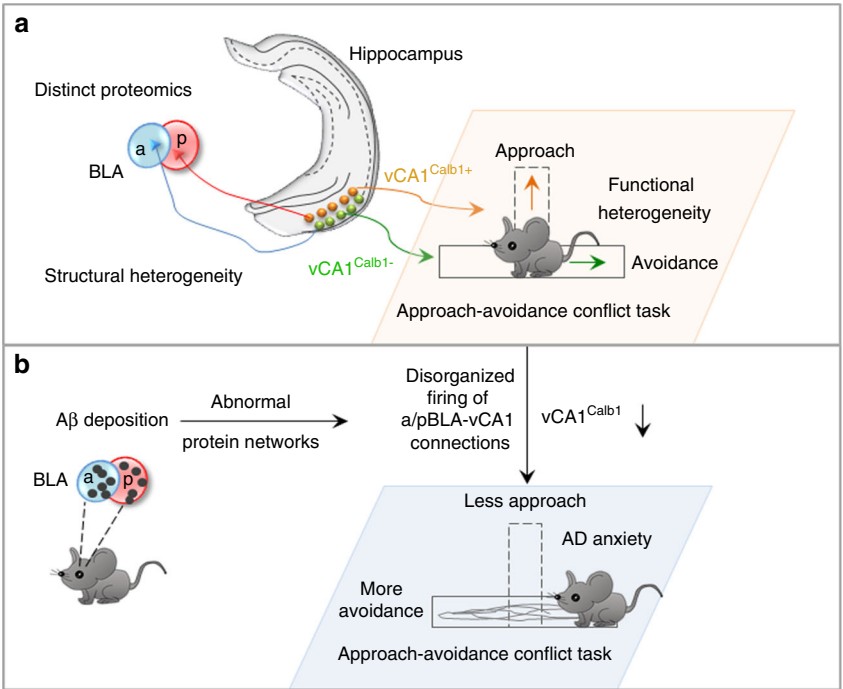

**Fig. 7 Proposed working model. a** Heterogeneity in the BLA–vCA1 circuit under physiological conditions. The BLA shows proteomic diversity along its anterior–posterior axis at the molecular level. The anterior part of the BLA (aBLA) and the posterior BLA (pBLA) innervate the deep-layer calbindin1-negative neurons (Calb1−) and superficial-layer calbindin1-positive neurons (Calb1+) in vCA1, forming aBLA–vCA1 and pBLA–vCA1 circuits, respectively. This molecular and structural heterogeneity endows these pathways with functional heterogeneity in controlling approach-avoidance behaviour, i.e., the aBLA–vCA1$^{Calb1-}$ circuit promotes avoidance and exerts an anxiogenic effect, while the pBLA–vCA1$^{Calb1+}$ circuit triggers approach and exerts an anxiolytic effect. **b** In AD, different protein network changes in response to Aβ deposition in the aBLA and pBLA impair the aBLA–vCA1$^{Calb1-}$ and pBLA–vCA1$^{Calb1+}$ circuits. Together with their disorganised firing patterns, the AD mice prefer avoidance over approach in conflict tasks and display anxiety. Furthermore, Calb1 expression determines the anxiolytic effect of pBLA–vCA1$^{Calb1+}$ stimulation in AD, indicating a molecular mechanism at the exit node of the pBLA–vCA1$^{Calb1+}$ circuit.

were robustly activated when animals were assessing aversive or safety information to make approach or avoidance decisions under conflict situations. The tonic stimulation of pBLA–vCA1 glutamic inputs or the brief optogenetic stimulation of vCA1$^{Calb1+}$ neurons when mice were approaching the open arm increased approach behaviour with decreased avoidance and retreat behaviours. In contrast to superficial vCA1$^{Calb1+}$ neurons, deep layer vCA1$^{Calb1-}$ neurons are the primary target of the aBLA. Furthermore, the deep layer of the vCA1 is enriched with anxiety cells, which represent anxiogenic stimuli and control avoidance behaviours[16]. Our data also confirmed that the optogenetic activation of aBLA–vCA1 inputs increased avoidance behaviour, while the inhibition of aBLA–vCA1 inputs decreased avoidance behaviour in conflict EPM and OFT paradigms. We speculate that anxiolytic and anxiogenic information from pBLA and aBLA may be respectively integrated in vCA1 and/or may be further transmitted to distinct downstreams of vCA1$^{Calb1+}$ and vCA1$^{Calb1-}$ neurons, by which they control whether and where to go or not to go when facing a complicated environment. Given that the excitatory neurons in the aBLA and pBLA could be telegraphed through mutual inhibition[19] and that Calb1+ neurons consist of excitatory neurons and GABAergic interneurons[31], it will be interesting in future studies to determine whether aBLA–vCA1 and pBLA–vCA1 circuits influence each other and how their coordination ultimately occurs in a conflict environment. In addition, the BLA shares reciprocal projections with the vCA1[32,33], and the vCA1–BLA circuit encodes contextual information in fear[33]. Thus, whether anxiolytic and/or anxiogenic information could be conveyed back to a/pBLA for

ultimate integration within the BLA–vCA1 loop circuit also needs further exploration.

Anxiety is an early symptom in AD patients and exacerbates memory deficits during the AD process[20,26]. Abnormal Aβ levels are increasingly believed to be responsible for both early cognitive (mild cognitive impairment) and affective symptoms (anxiety)[34,35]. Consistent with the findings in AD patients[25], we detected increased Aβ accumulation in the BLA of APP/PS1 mice compared with wild-type controls. We also observed the inhibition of the pBLA–vCA1 circuit in APP/PS1 mice, and photostimulation of pBLA–vCA1 terminals attenuated the circuit inhibition and increased approach behaviours. The pathological downward spiral theory implies that the fewer approach behaviours that occur, the fewer opportunities there are to learn that the situation may not be as horrible as expected, and this cycle can eventually induce anxiety. Thus, during the stimulation epoch, pBLA–vCA1 inputs can help recover normal avoidance behaviour in APP/PS1 mice with mechanisms involving promoting approach to regain the avoidance-approach balance. Stimulating pBLA–vCA1 inputs not only ameliorated anxiety but also improved spatial memory in APP/PS1 mice. These data support the results from a recent multi-center, prospective cohort study, which suggest that elevated anxiety symptoms can accelerate cognitive decline in preclinical AD[20,36]. Although the mechanism underlying anxiety-associated memory deficit is still unclear, our results highlight the key role of pBLA–vCA1 inputs in linking anxiety and memory impairment in AD. Our proteomic analyses also revealed many distinct biological processes between pBLA and aBLA, while the specific protein(s)/pathways contributing to the abnormal

activation pattern of aBLA–vCA1 and pBLA–vCA1 circuits may deserve further identification.

Calbindin1 functions as a buffer, sensor, and transporter of intracellular $Ca^{2+}$[37]. The Calb1 concentration varies in different types of hippocampal neurons[38]. As Calb1 can suppress free $Ca^{2+}$ increases and, thus, accelerate the collapse of the $Ca^{2+}$ gradient after the cessation of $Ca^{2+}$ influx[38,39], it is conceivable that calcium oscillation for information encoding could be very different between Calb1$^+$ neurons and Calb1$^-$ neurons during their excitation. We found that Calb1 knockdown in the vCA1 robustly abolished the anxiolytic effect of pBLA–vCA1 photostimulation in both wild-type and APP/PS1 mice, suggesting that Calb1 is crucial for neurons to encode assessment information about aversion and safety, thus controlling approach behaviours and exerting an anxiolytic effect. Therefore, both input specificity and Calb1 levels determine the pBLA–vCA1$^{Calb1+}$ circuit-associated amelioration of anxiety and memory deficits in AD mice. Given that there is currently no efficient therapy for AD but that the anxiety symptoms are amenable to treatment, our findings may also help inform risk stratification and optimise treatment for AD patients with anxiety and dementia.

In summary, we identified structural and functional heterogeneities in the BLA–vCA1 circuit. From its anxiogenic component, we dissected a novel anxiolytic projection from pBLA to vCA1$^{Calb1+}$ and demonstrated that the pBLA–vCA1$^{Calb1+}$ connection could control decision-making towards approach in conflict situations. The activation of the pBLA–vCA1$^{Calb1+}$ circuit significantly increased approach and reduced avoidance to exert an anxiolytic effect. In AD mice, the stimulation of the pBLA–vCA1$^{Calb1+}$ circuit not only ameliorated anxiety but also improved cognitive capacity. Thus, targeting the pBLA–vCA1$^{Calb1+}$ circuit could be a promising approach for anxiety disorders, such as AD.

## Methods

**Animals**. Adult male C57BL/6 mice (p45-60) were purchased from Beijing Vital River Laboratory Animal Technology Co., Ltd. Male APP/PS1 mice (APPswe, PSEN1dE9 and 85Dbo/MmJNju mice) were purchased from Model Animal Research Center of Nanjing University (Nanjing, China). Calb1-IRES2-Cre-D (B6. Cg-Calb1$^{tm2.1(cre)Hze}$/J,Jax. No. 028532) mice were purchased from Jackson Laboratory (USA) and were crossed with the tdTomato reporter Ai9 (Jax No. 007905) to fluorescently label Calb1$^+$ neurons. The animals were housed in groups of four to five per cage and were housed under a 12-h light/dark cycle (lights on at 6:00 p.m., off at 6:00 a.m.) at a stable temperature (23–25 °C). Food and water were given ad libitum. In the present study, we have complied with all relevant ethical regulations for the animal testing and research. All procedures were approved by institutional guidelines and the Animal Care and Use Committee (Huazhong University of Science and Technology, Wuhan, China) of the university's animal core facility.

**Anterograde tracing**. Mice were anaesthetised with 1.5% isoflurane at an oxygen flow rate of 1 L/min and head-fixed in a stereotactic frame (Stereotaxic for Mouse, SGL M,68030 Adaptor Inci; RWD Life Science Co., Ltd.). Eyes were coated with an erythromycin ointment, and body temperature was maintained with a heat lamp.

For anterograde polysynaptic tracing, H129-G4 or H129-R4 (100 nl, 2–5 × 10$^9$ pfu ml$^{-1}$) virus was injected into the aBLA (−1.34 AP, ±3.35–3.4 ML and −4.8 DV) or the pBLA (−2.3 AP, ±3.4 ML and −4.85 DV). One or two days later, mice were anaesthetised with a lethal dose of sodium pentobarbital and were perfused transcardially with ice-cold 0.1 M phosphate-buffered saline (PBS) followed by 4% paraformaldehyde. The expression of GFP or mCherry in the hippocampus was detected with a confocal microscope (LSM 780, ZEISS, Germany).

For anterograde monosynaptic tracing, a mixed helper virus containing AAV-EF1a-DIO-TK-GFP (3–6 × 10$^{12}$ vg ml$^{-1}$) and AAV-CaMKIIa-EGFP-P2A-Cre (2.5–5 × 10$^{12}$ vg ml$^{-1}$; 1:1, 150 nl) was delivered into the aBLA or pBLA of C57BL/ 6 wild-type mice. Three weeks later, H129-ΔTK-tdT (2–5 × 10$^8$ pfu ml$^{-1}$, 100 nl) was injected into the same site of aBLA or pBLA. Approximately 5 days later, the mice were killed with a lethal dose of sodium pentobarbital. The expression of tdTomato was observed in vCA1 along its superficial-deep axis by a fully automatic slice scanning system (OLYMPUS, SV120).

**CTB retrograde tracing**. Cholera toxin subunit B (recombinant), Alexa Fluor-647 conjugate was obtained from Thermo Fisher (C-34778). CTB (200–300 nl) was delivered stereotaxically into vCA1 (AP: −3.28 mm, ML: ± 3.3 mm, DV: −4.6 mm).

Seven days later, the mice were anaesthetised and perfused as what did in ante-rograde tracing. The expression of CTB in aBLA and pBLA was detected with a confocal microscope (LSM 780, ZEISS, Germany).

**shRNA**. To inhibit calbindin protein levels, we used AAV that was generated and purified by Obio Technology (Shanghai, China). The short hairpin RNA (shRNA) sequences targeting calbindin were 5′ -GCTGGATGCTTTGCTGAAAGA-3′. To achieve calbindin1 knockdown in vCA1 cells, a mixed virus containing AAV-CMV-bGlobin-Flex-EGFP-MIR30shRNA (Calb1) (2–5 × 10$^{12}$ vg ml$^{-1}$) and AAV-hSyn-EGFP-P2A-Cre (2.5–4.5 × 10$^{12}$ vg ml$^{-1}$; 1:1, 1 µl) was delivered into vCA1, and the control virus (AAV-CMV-bGlobin-Flex-EGFP-MIR30shRNA (NT), 2.5 × 10$^{12}$ viral vg ml$^{-1}$) was used. Approximately 1 month later, the mice were killed, and the brain protein was used for western blot.

**Patch-clamp electrophysiology**. The brains were sliced in ice-cold artificial cerebrospinal fluid (ACSF) containing (in mM): NaCl 124; KCl 3.0; MgCl$_2$ 1.0; CaCl$_2$ 2.0; NaH$_2$PO$_4$ 1.25; NaHCO$_3$ 26; glucose 10; and saturated with 95% O$_2$ and 5% CO$_2$ (pH 7.4). Then, the slices were incubated at 32 °C for 30 min in the same solution and allowed to equilibrate to room temperature for at least 30 min. Whole-cell patch-clamp recordings were made from visually identified superficial pyramidal neurons in the pyramidal layer of vCA1 after 5–6 weeks of AAV-CaMKIIa-hChR2(H134R)-EYFP infection in the pBLA. In current-clamp experiments, the recording electrodes (6–7 MΩ) were prepared on a P-97 puller (Sutter Instrument, Novato, CA) filled with (in mM): 135 potassium gluconate, 4 KCl, 2 NaCl, 10 HEPES, 4EGTA, 4 MgATP, 5 NaGTP, 280 mOsm kg$^{-1}$, pH adjusted to 7.4 with KOH. All recordings were made using a Multiclamp 700B amplifier (Molecular Devices, Sunnyvale, CA). Analogue signals were low-pass filtered at 1 kHz and digitised at 10 kHz using Digidata 1440 and pClamp9 software (Molecular Devices, Sunnyvale, CA). ACSF and drugs were applied to the slice via a peristaltic pump (Minipuls3; Gilson, Middleton, WI) at 2 ml min$^{-1}$. To activate pBLA–vCA1 terminals, square pulses of blue light (472 nm, 5 ms in duration) were delivered through a ×40 water-immersion objective of an Olympus microscope. LED served as a light source. The light power at the microscope objectives was ~2 mWmm$^{-2}$. Clampfit software (Molecular Devices, Sunnyvale, CA) was used for off-line analysis.

**Optogenetic manipulation in free-moving mice**. AAV5-CaMKIIa-eNpHR3.0-EYFP and AAV5-CaMKIIa-hChR2(H134R)-EYFP purchased from Brain VTA Technology Co., Ltd. Virus (150–200 nl, >10$^{12}$ vg ml$^{-1}$) was injected into aBLA (AP: −1.34 mm, ML:±3.4 mm, DV: −4.8 mm), pBLA (AP: −2.3 mm, ML: ± 3.45 mm, DV: −4.85 mm) and vCA1 (AP: −3.28 mm, ML: ±3.3 mm, DV: −4.6 mm) at a speed of 0.1 µl min$^{-1}$. Virus injection was counterbalanced across the left and right pBLA. Approximately 5 weeks later, optical fibres (core = 200 µm; numerical aperture = 0.37) were implanted in vCA1 (AP: −3.28 mm, ML:±3.3 mm, DV: −4.1 mm). Allowing 1 week for recovery, the mice then performed the behavioural tests.

**Elevated plus maze (EPM) test**. The EPM consists of two open arms (66 cm × 6 cm), two closed arms (66 cm × 6 cm) intersecting at 90 degrees in the form of a plus, with a central area (6 cm × 6 cm). The maze was elevated 50 cm from the floor. Mice were tested in a single 9-min session with three 3 min epochs. The test began with a light-off baseline epoch, followed by a light-on illumination epoch, and concluded with a second OFF epoch. During the light-on epoch, a constant yellow light (10 mW, 589 nm) or 20 Hz blue light (5 ms pulses, 5–8 mW, 472 nm) was delivered onto aBLA/pBLA–vCA1 terminals or vCA1$^{Calb1+}$ neurons through the optical fibres. The time the animal stayed in the open arm and the number of entries into the open arm was recorded. Between each trial, the maze was cleaned with 75% ethanol.

**Open field test (OFT)**. The open field chamber was made of transparent plastic (50 cm × 50 cm × 40 cm, length × width × height) and divided into a central field (25 cm × 25 cm) and a peripheral field. The central zone area was defined as 50% of the open field arena. Individual mice were placed in the centre of the chamber before starting the session. The 9-min session was the same as that in EPM. The time spent in the central area, the distance moved per min, and the moving speed were recorded by video-tracking and behavioural analysis software (Chengdu Techman Software Co., Ltd).

**L-maze test**. The L-maze consists of one open arm (30 cm × 6 cm) and one closed arm (30 cm × 6 cm) at a 90° angle in L shape. Compared with the EPM in which the mice have multiple choices, such as going back, going to the opposite arm, turning left, turning right and making transitory switches among these directions, the L-maze paradigm only provides two choices for the mice when they leave the transition zone, i.e., going to the opposite arm or going back to the previous arm. Obviously, the L-maze could be more helpful than EPM for the investigators to accurately determine the animals' decision-making behaviours. Therefore, we invented the L-maze for the optogenetic manipulation studies with vCA1$^{Calb1+}$ mice. When the mice shifted from the open arm to the transition zone (6 cm × 6 cm) or from the closed arm to the transition zone, an excitatory photostimulation (2 s, 8 mW, 472 nm) was delivered

onto vCA1$^{Calb1+}$ neurons. Mice were required to stay in the transition zone at least 0.5 s to receive a photostimulation. Control trials in which mice received no stimulation when they entered the transition zone were interspersed with stimulation trials in a pseudo-random order. A total of 20 trials were performed, 10 each of closed or open arm to transition zone stimulation trials and closed or open arm to transition zone no stimulation trials.

**Barnes maze (BM) test**. Different patterns and shapes of paper were placed on the walls of the behaviour testing room. One day before starting the training trials, mice were acclimatised in the target box for 3 min. Across the next 4 days of learning trials, three trials were performed each day with a 25-min interval. The animals were individually placed in a cylindrical chamber in the centre of the maze for 10 s and then trained to find the target hole within 3 min. If the animal failed to find the target hole at the end of each trial, they were directed to the target hole and left in the escape box for 1 min. Between each trial, the maze and the escape box were cleaned with 70% ethanol and a paper towel soaked with water. On day 5, mice were allowed 90 s to search in the BM without an escape box. During the learning trial, the delay to the target hole and the number of errors were recorded by the video-tracking software. During the probe trial, the percentage of correct pokes and the distance moved were measured.

**Fibre photometry in the EPM**. The change in neuronal activity in the EPM was assayed by recording GCaMP fluorescent signals with an optical fibre recording system (Thinker Tech Nanjing Biotech Limited Co., Ltd). Five weeks after AAV-EF1a-DIO-GCaMP6f virus was injected, an optical ceramic needle was inserted towards the vCA1, aBLA or pBLA through the craniotomy. The mice were housed individually for 7 days for recovery. A 488 nm laser (0.01–0.02 mW) was delivered using an optical fibre recording system, and fluorescent signals were recorded. For data analysis, the original signal is demodulated and converted to df/f. First, the raw signal is demodulated to return power at a frequency of 50 Hz. Next, the demodulated signal is converted to df/f using an average of 30 s before and after each data point as f, normalising each data point $f_n$ with the formula $(f_n − f)/f$. We set the time point at which the mouse entered the closed or open arm as 0. Then, we chose three specific behavioural events for df/f analysis: baseline (a 5-s period beginning 15 s before entering open/closed arm), pre-entry (a 5-s interval from beginning 8 s before entering open/closed arm) and entry (a 1 s after entering open/closed arm). Normalised df/f could monitor the activity alterations in the a/pBLA–vCA1 connection or Calb+ neurons stepwise along with decision priming and completion. Motion tracking and manual tagging were used to monitor and mark the position of the animals when they were exploring in the EPM.

**iTRAQ proteome experiment**. We performed aBLA and pBLA dissection and conducted the iTRAQ proteome experiment in double-blind conditions. Briefly, the aBLA and pBLA were excised and snap-frozen in liquid nitrogen. After tissue lysis, aBLA and pBLA were digested (200 μg for each sample). Peptides were labelled with TMT reagents according to the manufacturer's instructions (Thermo Fisher Scientific). Each aliquot (100 μg of peptide equivalent) was reacted with one tube of TMT reagent. After the sample was dissolved in 100 μL of 0.05 M TEAB solution, pH 8.5, the TMT reagent was dissolved in 41 μL of anhydrous acetonitrile. The mixture was incubated at room temperature for 1 h. Then, 8 μL of 5% hydroxylamine was added to the sample and incubated for 15 min to quench the reaction. The multiplex-labelled samples were pooled together and lyophilised. The TMT-labelled peptide mixture was fractionated using a Waters XBridge BEH130 column (C18, 3.5 μm, 2.1 × 150 mm) on an Agilent 1290 HPLC operating at 0.3 mL/min. The fractions were collected for each peptide mixture and then concatenated (pooling equal interval RPLC fractions). The fractions were dried for nano-LC-MS/MS analysis. LC-MS analysis was performed on a Q Exactive mass spectrometer that was coupled to Easy nLC (Thermo Fisher Scientific). The resulting LC-MS/MS raw files were imported into MaxQuant software (version 1.6.0.16) for data interpretation and protein identification against the database UniProt_Hordeum-vulgare_201747–20180125 (downloaded on 25/01/2018, including 201747 protein sequences), which is sourced from the protein database at https://www.uniprot.org/uniprot/?query=Hordeum+vulgare&sort=score. To reduce false positive identification results, a minimum unused score of 1.3 (equivalent to 95% confidence) and false discovery rate (FDR) <1% were required for all reported proteins. Finally, analyses of bioinformatics data were carried out with Perseus software, Microsoft Excel and R statistical computing software. Differentially significant expressed proteins were screened with the cut-off of a ratio fold-change of >1.20 or <0.83 and P values <0.05. GO enrichment analyses were carried out with Fisher's exact test, and FDR correction for multiple testing was also performed.

**Western blotting**. The brains were removed, and vCA1 was carefully dissected. The proteins were separated by SDS-polyacrylamide gel electrophoresis and probed with antibodies against polyclonal rabbit Calbindin1 (1:1000; Cat no. ab11426, Abcam) and polyclonal rabbit β-Actin (1:1000; Cat no. AC026, ABclonal). The blots were developed with horseradish peroxidase-conjugated secondary antibodies and visualised by an enhanced chemiluminescence substrate system (Santa Cruz Biotechnology Inc., Santa Cruz, CA, USA). The protein bands were quantitatively analysed by ImageJ.

**Immunofluorescence staining**. Mice were sacrificed 90 min after the last trial of the EPM by a lethal dose of sodium pentobarbital. After transcardial perfusion, the brains were removed and post-fixed in 4% paraformaldehyde overnight, after which they were transferred to a 25–30% sucrose solution in PBS for 3 days and sliced in 40 μm thick coronal sections. Sections were washed with PBS-T (PBS containing 0.1% Triton X-100) and subsequently incubated with polyclonal rabbit Calb1 (1:300; Cat no. ab11426, Abcam), goat c-Fos (1:200; Cat no. sc-52-G, Santa Cruz) or monoclonal mouse 6E10 (1:300; Cat no. 803001, BioLegend) for 17–20 h in QuickBlock™ Primary Antibody Dilution Buffer for Immunol Staining (P0262). After that, the sections underwent PBS-T washes (three times, 10 min each), followed by 1-h incubation with the secondary antibody (1:500, Cat no. A-21206, A-10040, A-11055 or A-21202 Invitrogen) at 37 °C. Finally, the slice underwent three more washes and counterstained with DAPI.

**Statistical analyses**. The commercial software (GraphPad Prism version 7; GraphPad Software, Inc, La Jolla, CA) were used for statistical comparisons, via one-way ANOVAs, two-way repeated ANOVAs and t tests to determine the different means among the groups. The significance threshold was set at $P = 0.05$, and the data were shown as mean ± SEM.

**Reporting summary**. Further information on research design is available in the Nature Research Reporting Summary linked to this article.

## Data availability

All data generated during this study are included in this published article. The source data underlying Figs. 1–4 and 6 and Supplementary Figs. 1–3, 6–10, 12 and 14–17 are provided as a Source Data file. The proteomics raw data have been deposited under the accession code PXD016515.

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

## Acknowledgements

This work was supported in parts by Natural Science Foundation of China (91632305, 91632111, 31730035, and 81721005), Ministry of Science and Technology of China (2016YFC1305801), Hubei Provincial Health and Family Planning Commission Youth Project to Y.Y (WJ2017Q014), Guangdong Provincial Key S&T Programme (2018B030336001) and NIH grant 1RF1MH120020-01. We thank Shanghai Bioprofile Technology Company Ltd for technological assistance in iTRAQ proteome experiment.

## Author contributions

This study was initiated and designed by J.Z.W. and Y.Y. J.Z.W. directed and coordinated the study. G.P. and D.G. performed major animal behaviour studies, virus injections and tracing, optogenetic manipulations. R.X. performed part of anterograde tracing. G.P. recorded the population calcium signalling and conducted the iTRAQ proteome experiment. D.W., Y.W., H.L. and H.Y. performed brain slice electrophysiology recordings, collected and analysed the data in double-blind way. Y.G., T.J., S.L., X.W. and T.H performed part of immunohistochemistry and western blotting. J.G., S.Z., T.Y. D.K. and R.L conducted part of virus injections and animal behaviour studies. D.K. performed genotyping. Y.Y., H.L. and G.L. performed imaging analyses. Y.Y. and X.Y. analysed the behaviour data in double-blind way. W.Z and M.H.L. supplied anterograde tracing virus. X.Z. helped to interpret the results and commented on the manuscript. Y.Y. and J.Z.W. wrote the manuscript.

## Competing interests

The authors declare no competing interests.
