## [Peer Review File · Nature Communications]

Reviewers' comments:

Reviewer #1 (Remarks to the Author):

This paper by Pi and colleagues seeks to elucidate a potentially important heterogeneity in the BLA-vHPC circuit related to approach/avoidance behavior building off previous work from their lab and others. Their novel finding that anterior and posterior BLA-vHPC projections differentially modulate behavior in the EPM and other tests is an interesting, potentially clinically relevant addition to the growing literature on how related circuitry might be manipulating these behaviors. However, their central finding that pBLA and aBLA-vHPC projectors are separate populations with opposite effects on approach/avoidance behavior is at odds with multiple papers they cite in recent literature. They discuss some of these findings, but often incorrectly interpret the previous results. They also fail to provide sufficient histological evidence to demonstrate the specificity of their injections for any of the experiments. These issues and others listed below should be addressed:

- o The paper was very hard to read and needs to be thoroughly rewritten. Far too many confusing or incorrect sentences to address each one here. This paper should be thoroughly edited for language.
- o The authors should show a detailed accounting of how many cells are in the AP spread of BLA as well as the spread of the projections in vHPC. Specifically, their previous paper as well as studies from other lab show there are very few aBLA-vHPC projecting cells. This raises the question of how comparable the projections are in terms of number, density, and topology.
- o A number of the papers they cite either broadly targeted BLA or vHPC and see the same optogenetic effects as they claim to for the aBLA projectors. Again given that multiple previous papers have claimed that aBLA is a far less significant population than the pBLA, it seems more likely that the previous results would have been primarily inhibiting the pBLA projectors, and that is what is suggested by the histology shown in some of those papers, particularly the Felix-Ortiz et al. 2013 paper. This needs to be properly addressed for both the optogenetic experiments and the calcium imaging.
- o Many of the experiments are underpowered with inappropriate statistics. This again is especially true for the patching and behavioral APP/PS1 experiments.
- o The authors repeatedly generalize their findings in troubling ways, most of which might be due to their issues with the losing clarity in the writing, but needs to be properly addressed in the edits.
 - ♣ For example, "Although negative and positive emotion were respectively elicited after aBLA and pBLA manipulation..."
 - ♣ Also, their claim in figure 7 about the relative strengths of the two projection patterns is not supported by their data
 - It's not clear that the second half of the paper adds much to the narrative and is a bit hard to interpret. To make sense of the shRNA manipulation before doing this test in the APP/PS1 mice they should've done a control in their WT cohorts to see if the shRNA injection has the same effect on the optogenetic anxiogenic and anxiolytic effects.
 - They should use different stats and change how they baseline for the EPM experiments, see Jimenez et al. 2018 and the Gunaydin et al. 2014 paper. They should show the histology for the imaging experiments, including co-labeling of their Calb+ line with a Calb+ antibody.
 - SuppFig8 is confusingly labeled, but is also not a convincing control for the level of initial presence of Calb+ cells and the change they see with stimulation. This approach doesn't control for the change in calbindin expression, which should be the outcome measure rather than the overlap with cFos. Obviously cells will have higher cFos after you stimulate them and you're more likely to see overlaps with anything else you stain for, but if they see the same increase in the actual calbindin concentration or can otherwise show a change in plasticity in the circuit that would be more persuasive.
 - Lastly, while previous papers used different methods to look at BLA-vHPC projectors and also the general activity of vHPC CA1 cells in approach/avoidance and other tasks, they have not reported the type of clear spatial segregation of encoding along the superficial-deep axis that the optogenetic and calcium imaging results in this paper suggest would be present. This again needs

to be directly addressed.

Reviewer #2 (Remarks to the Author):

The manuscript tests the hypothesis that BLA projections to the ventral HPC are composed of distinct pathways - an anterior input to the deep HPC and a posterior input the superficial HPC. The authors provide substantial evidence that these two projections form unique roles in anxiety-like behavior in rodents. In addition, the authors propose a novel hypothesis regarding Alzheimer's disease and anxiety-like behavior reported in rodent APP models. The manuscript is well written, however several methodological details are missing from the manuscript. My majors comments are listed below.

1. It is a bit difficult to determine which region of vCA1 is included in Figure 1. Could the authors provide a low magnification image and with a box indicating where the higher magnification image was taken? Also, did the projections extend into intermediate HPC?
2. Figure 1 data - A definition for the superficial-deep axis of vCA1 should be provided in the Methods section or in the results section where it is first discussed. This could also be illustrated in a schematic and presented early in the results section or perhaps as a panel in Figure 1.
3. OFT - The approximate size of the central zone should be described.
4. In Figure 4, Ca imaging data is presented showing an increase in overall fluorescence when entry into the closed arm is available. On the other hand, overall fluorescence decreases when entry into the open arm is made available. The relationship between population Ca fluorescence is only correlated with time (as shown in Fig. 4). Does time represent the animals entry into the open/closed arms or does it represent when the doors to the arms were opened? Was motion tracking used to monitor the position of the animal as it entered the arms - was this done manually with video records or using tracking software? It is unclear how the neural activity was correlated with the animals behavior.
5. Some additional details regarding the relationship between Ca transients and the animals behavior is warranted. Did the authors examine the locomotor speed of the animal when they transitioned between open/closed arms? Perhaps differences in locomotor variables are driving this population response as shown in Fig 4 and 6. Further, it is possible that locomotor differences between APP and WT animals are driving the different Ca populations responses shown in Fig. 6.

Reviewer #3 (Remarks to the Author):

The authors report that projections from the anterior and posterior basolateral amygdala (aBLA and pBLA, respectively) to the ventral hippocampus CA1 (vCA1) have opposite functions. Activation of pBLA-vCA1 and aBLA-vCA1 respectively decrease and increase anxiety in mice. Furthermore, the authors show that pBLA targets calb1+ neurons, while aBLA targets calb1- neurons. Photometry recordings show that calb1+ vCA1 neurons are activated when mice enter the safe closed arms of the maze in normal mice, but not in a model of Alzheimer's Disorder (AD). AD mice also show distinct transcriptional changes in aBLA and pBLA compared to control mice. Lastly, AD mice show increase anxiety and lower performance in the Barnes Maze. Both of these deficits can be rescued by optogenetic activation of pBLA-vCA1 in control mice, but not in mice that show reduced levels of calb1 in vCA1.

The main novelty in this paper is showing that aBLA and pBLA projections to the hippocampus

activate non-overlapping and molecularly distinct (calb1+ and calb1- cells) populations, and that these projections influence anxiety in opposing ways. These findings are important and of interest to the general systems neuroscience community. However, there are several points that must be addressed prior to publication.

1. The authors often use incorrect and awkward phrasing. Sometimes this problem obscures what the authors want to convey. There are dozens of occurrences of grammatical errors, such as:
“inputs with disordered firings and heterogenized protein networks”
“the decreased approaching exploration renders them failed to extinct the previous fear”
“ventral hippocampal CA1 (vCA1) is identified as anxiety cell niche”
This article cannot be published unless it the writing is thoroughly corrected by a neuroscientist that can write well in English.

2. Calbindin expressing cells in the hippocampus can be found both amongst pyramidal cells (Kohara et al., 2014), as well as GABAergic interneurons in CA1 (Jinno and Kosaka, 2006). The authors should show if the calb1+ cells that receive pBLA inputs are gabaergic or glutamatergic. If they are gabaergic, the authors could show if these cells inhibit the calb1- cells that receive aBLA input. These data would provide a very nice mechanistic explanation for the opposing functions of the aBLA and pBLA input to vCA1.

3. The authors repeatedly state that calb1+ vCA1 cells “encode “secure” information”. This statement implies that these cells are constantly activated during exposure to safe locations. However, their own data in Fig. 4b and 4e shows that these cells are activated during the entry to the closed arm, not during the whole time the mouse is in the closed arm. The authors should change the text throughout the manuscript and state instead that these cells are activated immediately prior to entry into the closed arm.

4. I haven't checked the references throughout the paper, but I found a few incorrect citations. For example: “theta–frequency firing within ventral hippocampus (vHPC) is synchronized with medial prefrontal cortex (mPFC) discharge during exposure to the anxiogenic environments^{2,14}”
References 2 and 14 do not have any data related to theta-frequency firing. Please carefully check if all citations are correct.

5. The authors state that “L–type maze has definite moving direction which can help the investigators to accurately determine the animals' decision–making behaviors.” It is not clear from this sentence why the authors use the L-maze instead of the more common plus maze. Please clarify.

6. The authors wrote that “stimulation of pBLA–vCA1 inputs in APP/PS1 mice may break the vicious cycle to ameliorate AD anxiety by promoting approach to regain the avoidance–approach balance”. This statement implies that temporary stimulation of pBLA–vCA1 inputs would have some long-term effect to increase approach, but the authors have not done any experiments to show this. The data in Fig 2 actually show that after stimulation is turned off the animals resume normal avoidance, without any long-term stimulation effect.

7. The authors should write a short sentence describing why they chose APP/PS1 mice as an AD model.

8. It is unclear how the authors are defining pBLA. For example, Fig 2a and 2g both show pBLA, but the photographs show sections that are in very different positions in the anterior-posterior axis.

9. In Fig 1a the authors use viruses to perform anterograde monosynaptic tracing. While this method has been used before, it has not been used extensively and validated by many researchers. Please show in acute slice experiments that anterogradely tagged cells respond to optogenetic stimulation of BLA inputs while non-tagged cells in the same layer do not respond to stimulation of BLA inputs.

Please find below a point-by-point reply (in black) to the reviewers' comments (in blue):

Reviewer #1 (Remarks to the Author):

This paper by Pi and colleagues seeks to elucidate a potentially important heterogeneity in the BLA–vHPC circuit related to approach/avoidance behavior building off previous work from their lab and others. Their novel finding that anterior and posterior BLA–vHPC projections differentially modulate behavior in the EPM and other tests is an interesting, potentially clinically relevant addition to the growing literature on how related circuitry might be manipulating these behaviors. However, their central finding that pBLA and aBLA–vHPC projectors are separate populations with opposite effects on approach/avoidance behavior is at odds with multiple papers they cite in recent literature. They discuss some of these findings, but often incorrectly interpret the previous results. They also fail to provide sufficient histological evidence to demonstrate the specificity of their injections for any of the experiments. These issues and others listed below should be addressed:

[Response]: First of all, we greatly appreciate the reviewer for the positive evaluation of our novel findings and the significance of the paper.

We deeply regret for our mistakes in reference insertion. We have now very carefully checked the citations and made corrections accordingly. By analyzing the coordinates of mouse brain atlas, we found that most of the previous works on BLA mainly targeted anterior BLA (aBLA) not posterior BLA (pBLA) or whole BLA (a/pBLA). Thus, we have corrected the inappropriate interpretation of the previous results in the introduction part of main text. For example, we corrected sentence “Non-specific activation of glutamatergic somata elicits anxiogenic effect, while stimulation of its terminals in the central nucleus of amygdala (CeL) shifts anxiogenic to anxiolytic effect” into “Non-specific activation of glutamatergic somata of BLA, especial in its anterior part (aBLA), elicits anxiogenic effect. While selective stimulation on its terminals in central nucleus of amygdala (CeA) or the anterodorsal part of bed nucleus of stria terminalis (adBNST)

shift anxiogenic to anxiolytic.” (page 4, line 58 to 61). We also corrected the inappropriate interpretation of the previous vCA1 results, such as changing “By *in vivo* Ca²⁺ imaging, ventral hippocampal CA1 (vCA1) is identified as anxiety cell niche to represent anxiety-related information, and optogenetic inhibition of vCA1 anxiety cell reduces avoidance behavior to produce anxiolytic effect¹⁶.” into “By *in vivo* Ca²⁺ imaging, anxiety cells were discovered in ventral hippocampal CA1 (vCA1). Optogenetic inhibition of vCA1 anxiety cells reduces avoidance behavior to produce anxiolytic effect.” (please see page 4 line 66-68).

After correcting the misleading citations and improving the poor writing, it can be seen that the data presented in our current study are not contradictory to the previous reports, instead, these novel findings are actually in good agreement with and nicely complementary to the previous reports. For instance, our findings of anxiogenic effect of aBLA–vCA1 connection are in good agreement with previous works (Tye et al., 2011 and Felix–Ortiz et al., 2013). Jimenez et al. reported that anxiety cells were distributed in the deep layer of vCA1 pyramidal cell (PC) and their activation produced anxiety-related behaviors. We found in the present study that aBLA dominantly innervated the deep layer of vCA1 PC. Taken together, those findings may consistently explain why aBLA–vCA1 connection is anxiogenic. Since pBLA–vCA1 circuit had not been studied in anxiety-related behaviors, we employed neural circuit tracing, optogenetics and *in vivo* calcium recording to address this issue. We found that pBLA preferentially innervated Calb1+ neurons in the superficial vCA1 PC layer and pBLA–vCA1 connection controlled approach behavior to exert anxiolytic effect with Calb1–dependent manner. These distinct innervation pattern and anxiolytic effect of pBLA–vCA1 circuit are the central novelty of our current work and should be viewed as an essential supplement to the studies of a/pBLA–vCA1 connections, rather than the conflict results of aBLA–vCA1 inputs studies. In the revised version, this matter has been emphasized throughout the manuscript.

They also fail to provide sufficient histological evidence to demonstrate the specificity of their injections for any of the experiments.

[Response]: We agree with the reviewer that histological evidence of virus injection sites is important to demonstrate the targeting manipulation. In the previous version of the manuscript, we did show the injection sites of AAV5–CaMKIIa–hChR2(H134R)–EYFP and AAV5–CaMKIIa–eNpHR3.0–EYFP in Figure 2 (a, g) and Figure 3 (a, g). From these figures, we could see that the expression of eYFP was respectively confined to the aBLA and pBLA subregions after the injection, which provides evidence for the precise targeting on aBLA and pBLA.

To provide further histological evidence for the specificity of the injections as suggested by the reviewer, we have done additional experiments by series slicing of BLA after anterograde multisynaptic tracing (please see sFigure 1). Robust GFP signals were detected respectively in the aBLA (sFigure 1a) and pBLA (sFigure 1f) after their targeting injections, which confirmed the accuracy of the injection. Furthermore, distinct innervating patterns from aBLA and pBLA to the deep and superficial layer of vCA1 were also shown (sFigure 1c, d, g, h). In addition to vCA1, we also found other robust projecting signals originated from aBLA and pBLA. At bregma -1.82 mm, -2.3 mm and -3.4 mm, we detected GFP signals in central nucleus of the amygdala (CeA), dorsal endopiriform nucleus (Den), piriform cortex (Pir), perirhinal cortex (PRh), entorhinal cortex (Ect), temporal association cortex (TeA), primary auditory cortex (Au1), secondary auditory cortex (AuV), intermediodorsal thalamic nucleus (IMD), posterior part of basomedial amygdaloid nucleus (BMP), anterolateral part of amygdalohippocampal area (AHiAL), posterolateral cortical amygdaloid nucleus (PLCo), lateral entorhinal cortex (Lent), dorsal terminal nucleus of the accessory optic tract (DT), posterior intralaminar thalamic nucleus (PIL) peripeduncular nucleus (PP), lateral entorhinal cortex (Lent) and amygdalopiriform transition area (APir) (sFigure 1e–g). By literature searching, we found that BLA (actually aBLA according to the atlas)–CeA^{1, 2, 3}, LA/BLA–Pir⁴, and BLA (actually a/pBLA according to the atlas)–Lent⁵ projections had been identified by different tracking methods. Our novel findings on connections between aBLA and pBLA, and the direct or indirect projections from aBLA or pBLA to Den, PRh, TeA, Au1, AuV, IMD, BMP, AHiAL, PLCo, DT, PIL,

Ect, APir, and Lent have provided supplementary information in a/pBLA-projecting circuits and deserve for further investigation. We have added all these new data as sFigure 1.

Supplementary Figure 1. Trans-synaptic virus tracing a/pBLA-vCA1 circuits. (a–g) Anterograde H129–G4 was injected into the aBLA (a) and pBLA (f), respectively. 24 h later, mice were sacrificed for series slicing. Representative images of aBLA–(a–c) and pBLA–projecting (e–g) regions by serial coronal sections. Scale bar=1 mm. (d, h) Higher-resolution images of the boxed regions in panel c and g, showing that H129–G4 injected in aBLA preferentially label the deep layer of vCA1 PCs (d), while in pBLA predominantly label the superficial layer of vCA1 PCs (h). Scale bar=100 μ m. (i–l) Comparison of the projection strength of aBLA and pBLA to vCA1: (i) Quantitative analysis of the starter cells from anterior (A) to posterior (P) axis in aBLA–H129 and pBLA–H129 mice, respectively. Note that the maximum values of the two groups (aBLA–H129 group at bregma -1.46 vs pBLA–H129 group at bregma -2.46) was identical. $n=7$ per group, unpaired t test, $t=1.842$ $df=11.99$, $P=0.0903$. (j) Comparison of the number of H129–labeled vCA1 cells in aBLA–H129 and pBLA–H129 mice, showing that more vCA1 neurons were innervated by pBLA than aBLA from bregma -3.08 to -3.64. $n=7$ per group, two-way ANOVA, $F(6, 72) = 5.103$, $P=0.0002$, Sidak's multiple comparisons test, $*P < 0.05$, $**P < 0.01$ vs pBLA–H129. (k, l) Comparison of the number of deep (k) and superficial (l) vCA1 cells labeled by H129 in aBLA–H129 and pBLA–H129 mice. From bregma -2.92 to -3.64, more H129–labeled cells were located in the deep layer of vCA1 PCs in aBLA–H129 mice as compared with those in pBLA–H129 mice (k). While the number of H129–labeled superficial vCA1 cells was greater in pBLA–H129 mice than those in aBLA–H129 mice (l). $n=7$ per group, Two-way ANOVA, deep layer of vCA1 PCs: $F(6, 72) = 6.96$, $P<0.0001$; superficial layer of vCA1 PCs: $F(6, 72) = 22.41$, $P<0.0001$, Sidak's multiple comparisons test, $**P < 0.01$ vs pBLA–H129 (k) or aBLA–H129 (l). so, stratum oriens; sp, stratum pyramidale; sr, stratum radiatum.

In addition to anterograde multisynaptic tracing, we also employed H129– Δ TK–tdT and helper virus to outline the monosynaptic connection between a/pBLA and vCA1

(Figure 1a). Although these viruses perform well in monosynaptic mapping, one limitation in this field is that the TK complementation leads to severe cell damage and cytopathy of starter cells in the injection site⁶. Thus, we were unable to present histological evidence of virus injection sites at day 7 after H129- Δ TK-tdT injection due to the technique limitation.

o The paper was very hard to read and needs to be thoroughly rewritten. Far too many confusing or incorrect sentences to address each one here. This paper should be thoroughly edited for language.

[Response]: We are very sorry for the poor writing of the manuscript. As suggested, we have extensively revised the manuscript (highlighted in red) and asked professional editing services for proofreading. We believe that the clarity of the paper has been substantially improved.

o The authors should show a detailed accounting of how many cells are in the AP spread of BLA as well as the spread of the projections in vHPC. Specifically, their previous paper as well as studies from other lab show there are very few aBLA-vHPC projecting cells. This raises the question of how comparable the projections are in terms of number, density, and topology.

[Response]: Thanks to the reviewer for the constructive comments. As suggested, we first counted the starter cells in the aBLA of aBLA-H129 mice and the pBLA of pBLA-H129 mice respectively from bregma -0.94 mm to -2.92 mm (sFigure 1i). Quantitative analysis showed that the highest abundance of H129-labeled aBLA cells was concentrated from bregma -1.34 mm to -1.82 mm in aBLA-H129 group. While, in pBLA-H129 group, most H129-labeled pBLA cells were distributed from bregma -2.18 to -2.54. The maximum values between the two groups (aBLA-H129 group at bregma -1.46 mm vs pBLA-H129 group at bregma -2.46 mm) had no difference (sFigure 1i). To compare their innervation strength on vCA1 in terms of number, we then counted H129-labeled vCA1 cells in aBLA-H129 and pBLA-H129 mice, respectively (sFigure

1j). We observed that the number of H129-labeled vCA1 cells was greater in pBLA-H129 mice than that in aBLA-H129 mice (sFigure 1j), indicating that more vCA1 cells were predominantly innervated by pBLA than by aBLA in physiological condition. Considering the difference of innervation pattern between aBLA-vCA1 and pBLA-vCA1 connections, we next compared the number of H129-labeled cells in the superficial and deep layers of vCA1 in aBLA-H129 and pBLA-H129 mice (sFigure 1k, l). In the deep layers of vCA1 PC, about 80 and 20 cells labeled by H129 in each section from bregma -2.92 to -3.64 were respectively detected in aBLA-H129 and pBLA-H129 mice (Figure 1k), indicating preponderant connections between aBLA and vCA1^{deep} over pBLA and vCA1^{deep}. However, in the superficial layers of vCA1 PC, ~20 and ~120 cells labeled by H129 in each section from bregma -2.92 to -3.64 were found in aBLA-H129 and pBLA-H129 mice (Figure 1l), suggesting an overwhelming superiority of pBLA-vCA1^{superficial} in a/pBLA-vCA1 connections. Now, we think our new results on H129 anterograde tracing have provided quantitative details of aBLA-vCA1 and pBLA-vCA1 connections along the AP axis of BLA and the superficial-deep axis of vCA1, as suggested by the reviewer.

Supplementary Figure 1. Trans-synaptic virus tracing a/pBLA-vCA1 circuits. (a-

g) Anterograde H129–G4 was injected into the aBLA (**a**) and pBLA (**f**), respectively. 24 h later, mice were sacrificed for series slicing. Representative images of aBLA–(**a**–**c**) and pBLA–projecting (**e**–**g**) regions by serial coronal sections. Scale bar=1 mm. (**d**, **h**) Higher–resolution images of the boxed regions in panel **c** and **g**, showing that H129–G4 injected in aBLA preferentially label the deep layer of vCA1 PCs (**d**), while in pBLA predominantly label the superficial layer of vCA1 PCs (**h**). Scale bar=100 μ m. (**i**–**l**) Comparison of the projection strength of aBLA and pBLA to vCA1: (**i**) Quantitative analysis of the starter cells from anterior (A) to posterior (P) axis in aBLA–H129 and pBLA–H129 mice, respectively. Note that the maximum values of the two groups (aBLA–H129 group at bregma -1.46 vs pBLA–H129 group at bregma -2.46) was identical. $n=7$ per group, unpaired t test, $t=1.842$ $df=11.99$, $P=0.0903$. (**j**) Comparison of the number of H129–labeled vCA1 cells in aBLA–H129 and pBLA–H129 mice, showing that more vCA1 neurons were innervated by pBLA than aBLA from bregma -3.08 to -3.64. $n=7$ per group, two–way ANOVA, $F(6, 72) = 5.103$, $P=0.0002$, Sidak's multiple comparisons test, $*P < 0.05$, $**P < 0.01$ vs pBLA–H129. (**k**, **l**) Comparison of the number of deep (**k**) and superficial (**l**) vCA1 cells labeled by H129 in aBLA–H129 and pBLA–H129 mice. From bregma -2.92 to -3.64, more H129–labeled cells were located in the deep layer of vCA1 PCs in aBLA–H129 mice as compared with those in pBLA–H129 mice (**k**). While the number of H129–labeled superficial vCA1 cells was greater in pBLA–H129 mice than those in aBLA–H129 mice (**l**). $n=7$ per group, Two–way ANOVA, deep layer of vCA1 PCs: $F(6, 72) = 6.96$, $P<0.0001$; superficial layer of vCA1 PCs: $F(6, 72) = 22.41$, $P<0.0001$, Sidak's multiple comparisons test, $**P < 0.01$ vs pBLA–H129 (**k**) or aBLA–H129 (**l**). so, stratum oriens; sp, stratum pyramidale; sr, stratum radiatum.

o A number of the papers they cite either broadly targeted BLA or vHPC and see the same optogenetic effects as they claim to for the aBLA projectors. Again given that multiple previous papers have claimed that aBLA is a far less significant population than the pBLA, it seems more likely that the previous results would have been primarily inhibiting the pBLA projectors, and that is what is suggested by the histology shown in some of those papers, particularly the Felix–Ortiz et al. 2013 paper. This needs to be properly addressed for both the optogenetic experiments and the calcium imaging.

[Response]: Thanks to the reviewer for the comments. We apologize for our incorrect interpretation of the previous results in the references. By carefully checking the coordinates in the papers of Tye et al., 2011³ and Felix–Ortiz et al. 2013⁷, we found that the injection sites “bregma: -1.6 mm anteroposterior, ±3.1 mm mediolateral, -4.5 mm dorsoventral” (Tye et al., 2011) and “bregma: -1.6 mm anteroposterior, ±3.2-3.4 mm mediolateral, -4.9 mm dorsoventral” (Felix–Ortiz et al. 2013) were precisely targeted aBLA not a/pBLA in mice. During photostimulation epoch, they found that aBLA and aBLA–vCA1 circuit exerted anxiogenic effect, which was also confirmed in our current study.

We totally agree with the reviewer that broad damage on BLA can inhibit pBLA and its projection to vCA1. However, the interpretation of final outcome should not be simply attributed to the relative strength between aBLA–vCA1 and pBLA–vCA1 connections, because BLA could orchestrate to modulate emotion *via* many downstreams. Previous papers have claimed that aBLA neurons send axons to vCA1⁷, BNST⁸, mPFC⁹, CeA^{1, 2, 3}, Pir⁴ and Lent⁵. Among them, at least aBLA–innervated vCA1⁷ and mPFC¹⁰ neurons had been proven to regulate anxiety–related behaviors, while aBLA–innervated BNST and CeA had been identified to exert anxiolytic effects^{3, 8} (please note that the above studies only used the term of BLA without further identifying the subregions, we have identified the aBLA according to their injection site). Thus, the final outcome of the manipulation on a/pBLA population neurons should depend on the convergence effect of all the downstreams of a/pBLA.

o Many of the experiments are underpowered with inappropriate statistics. This again is especially true for the patching and behavioral APP/PS1 experiments.

[Response]: Thanks to the reviewer for pointing out this important issue. We have increased number of experimental mice from 3 to 7 mice (13 neurons) in the patching experiments (please see Figure 1j) and from 3 to 10 and 11 mice per group in population calcium recording on APP/PS1 mice (please see Figure 6a–d). The same conclusions were confirmed.

o The authors repeatedly generalize their findings in troubling ways, most of which might be due to their issues with the losing clarity in the writing, but needs to be properly addressed in the edits.

[Response]: We are very sorry for the poor writing of the manuscript. As suggested, we have substantially revised the paper (highlighted in red) and asked professional editing services for proofreading of the manuscript. We believe that the clarity has been greatly improved by the revisions.

♣ For example, “Although negative and positive emotion were respectively elicited after aBLA and pBLA manipulation...”

[Response]: We revised the sentence into “Manipulating anterior BLA (aBLA) and posterior BLA (pBLA or BLP) can elicit respectively negative and positive emotional behaviors, but their heterogeneity at molecular level is still unclear.” (please see page 16 line 381 to 383).

♣ Also, their claim in figure 7 about the relative strengths of the two projection patterns is not supported by their data

[Response]: We appreciate the opportunity to clarify this point. In Figure 7, we proposed a working model to summarize the heterogeneities of BLA–vCA1 circuit under physiological and the AD conditions. Based on our current H129 anterograde (sFigure 1i-l) and CTB retrograde tracing data (Figure 6e), as well as our previous

report (Yang et al., 2016), we believe that the data on relative strengths of the two projections in physiological condition were solid, *i.e.*, the pBLA–vCA1 connection are stronger than aBLA–vCA1 connection in terms of the innervation strength and the degree of activation during elevated plus maze (EPM) test.

In AD mice, we observed that the CTB signals were remarkably decreased in both aBLA and pBLA, and the reduction was more significant in pBLA than that in aBLA (Figure 6e). By measuring the global activity of a/pBLA–vCA1 circuits in the EPM test using c-Fos staining combined with CTB tracing, we observed that the pBLA–vCA1 circuit in the AD mice was failed to be activated dominantly over aBLA–vCA1 circuit as observed in wild-type mice (Figure 6e). These data suggest that disruption of pBLA–vCA1 connection in the AD mice may screw the balance toward aBLA–vCA1 connection, but the absolute strength of pBLA–vCA1 connection does not necessarily become less than aBLA–vCA1 connection as we have proposed in the previous version of manuscript. Therefore, we have revised the expression of “aBLA–vCA1^{Calb1- >} pBLA–vCA1^{Calb1+>}” into “a/pBLA–vCA1 circuit imbalance” to accurately explain the strength shift between these two projections in the AD mice (please see Figure 7b).

[REDACTED]

Figure 7. The proposed working model. (a) Heterogeneity of BLA–vCA1 circuit in physiological condition. BLA has proteomic diversity along its anterior–posterior axis

in molecular level. The anterior part of BLA (aBLA) and the posterior BLA (pBLA) respectively innervate the deep layer calbindin1–negative neurons (Calb1-) and superficial layer calbindin1–positive neurons (Calb1+) in vCA1 forming aBLA–vCA1 and pBLA–vCA1 circuits. These molecular and structural heterogeneities endow their functional heterogeneities in controlling approach–avoidance behavior, *i.e.*, aBLA–vCA1^{Calb1-} circuit promotes avoidance exerting anxiogenic effect, while pBLA–vCA1^{Calb1+} circuit triggers approach playing anxiolytic role. **(b)** In AD, different protein network changes in response to A β deposition in the aBLA and pBLA impair the balance between aBLA–vCA1^{Calb1-} and pBLA–vCA1^{Calb1+} circuits. Together with their disorganized firing, the AD mice prefer avoidance than approach in conflict tasks and display anxiety. Furthermore, Calb1 expression determines the anxiolytic effect of pBLA–vCA1^{Calb1+} stimulation in AD, indicating molecular mechanism at exit node of pBLA–vCA1^{Calb1+} circuit.

• It's not clear that the second half of the paper adds much to the narrative and is a bit hard to interpret. To make sense of the shRNA manipulation before doing this test in the APP/PS1 mice they should've done a control in their WT cohorts to see if the shRNA injection has the same effect on the optogenetic anxiogenic and anxiolytic effects.

[Response]: As suggested by the reviewer, we performed shCalb1 experiment in wild-type cohorts. The results showed that knockdown Calb1 in vCA1 neurons exerted anxiogenic effect no matter pBLA–vCA1^{Calb1+} connection was activated or not (sFigure 14). These data strongly support that Calb1 is the molecular basis for the anxiolytic effects of pBLA–vCA1^{Calb1+} connection. We also extended the results part of this supplementary data on page 14, lines 326–334.

Supplementary Figure 14. Downregulating Calb1 in the vCA1 induces anxiogenic effects in mice.

(a) Representative image of the injection site in vCA1. Scale bar, 100 μ m. (b, c) Expression of Calb1 was significantly decreased after AAV–ShCalb1 injection as compared with AAV–ShNT group by Western blotting. $n=3$ per group. Unpaired t test, $t=8.338$, $df=4$, $P=0.0011$. (d, e) In EPM, shCalb1 mice showed less time staying in open arm (d) and probability of open–arm entry (e) as compared with shNT mice. Optogenetic activation of pBLA–vCA1 inputs increased the time (d) and the open–arm entry (e) in shNT mice but not in shCalb1 mice (One–way ANOVA, time in open arm: $F(3, 40)=32.86$, $P<0.0001$; probability of open arm entry: $F(3, 40)=48.09$, $P<0.0001$. (f) In OFT, shCalb1 mice showed significantly reduced staying time in the center field (f) as compared with shNT mice. Optogenetic activation of pBLA–vCA1 inputs increased the time (f) in shNT mice but not in shCalb1 mice. One–way ANOVA, $F(3, 40)=25.08$, $P<0.0001$. (g) Distance moved in OFT was not changed among the groups.

One-way ANOVA, $F(3, 40) = 0.03032$, $P = 0.9928$. $n = 11$ mice per group, $*P < 0.05$, $**P < 0.01$ vs shNT OFF group, $###P < 0.01$ vs shNT ON group. Data are presented as mean \pm SEM.

- They should use different stats and change how they baseline for the EPM experiments, see Jimenez et al. 2018 and the Gunaydin et al. 2014 paper. They should show the histology for the imaging experiments, including co-labeling of their Calb+ line with a Calb+ antibody.

[Response]: Thanks to the reviewer for the suggestion. To identify whether and how the activity of a/pBLA–vCA1 circuits correlate with approach–avoidance decision, we had set a short observation window in the process of EPM transition and used population calcium recording to delineate the temporal activity patterns of aBLA and pBLA neurons in a/pBLA–vCA1 circuits. Due to the lack of definite behavior signs during decision–making process in EPM except at decision–made moment, we hope the reviewer agrees that the baseline setting for social behavior (as reviewer mentioned in Gunaydin et al. 2014 paper) may not be suitable for our research on decision–making process in EPM. In another reference mentioned by the reviewer (Jimenez et al. 2018), the researchers used microendoscopic calcium imaging to visualize every individual neuron’s activity when mice were exploring the EPM. In this method, a minimum z–projection image of the entire movie was set as the reference F0 (baseline). After normalizing fluorescence signals of each neurons *via* $\Delta F/F_0$, researcher selected cells whose calcium event rates were different between zones (EMP: open–closed; OFT: center–periphery) for further analysis to define the anxiety–related cells. While, in the present study, our optical fiber recording system could only record the population calcium transits. Thus, the “neutral cells” whose calcium event rates were identical in open and closed arms might largely mask the calcium changing of our “approaching/avoidance–decision–making cells” in population calcium transits. Therefore, we hope the reviewer agrees that this method of baseline setting may not be suitable for our population calcium analysis.

As the population calcium signals were sensitive to some postures, such as headdip, stretch, climb and so on, when mice were exploring in EPM, the widely varied calcium transits could be hardly converted to df/f using a fixed period for f calculation and interpreted by comparison based on simple average calculation. However, rolling average in statistics is a good way to smooth out trends over a significant period of time by minimizing the interference of random variation in the raw data. Thus, we referenced LeBlanc' work (LeBlanc et al., 2018) and used rolling average to normalize each data point around transition zone including baseline (a 5 s period beginning 15 s before entering open/closed arm), pre-enter (a 5 s interval from beginning 8 s before entering open/closed arm) and enter phase (a 1 s after entering open/closed arm) to identify the "rolling" picture of population calcium signals as time progresses around transition zone in EPM. These normalized df/f could monitor the alteration of neuron activity step by step along with the decision priming and the completion. Thus, these data could provide convinced evidence to illustrate the relationship between the activity of a/pBLA neurons in a/pBLA-vCA1 circuits and the approach-avoidance decision in conflict condition. We have clarified these in the revised paper (please see page 24 lines 603-609).

To co-label the Calb⁺ line with Calb antibody as suggested by the reviewer, we have done additional experiments by crossing Calb1-IRES2-Cre-D (B6.Cg-Calb1^{tm2.1(cre)Hze}/J, Jax. No.028532) mice with tdTomato reporter Ai9 (Jax No. 007905) to fluorescently label Calb1⁺ neurons with tdTomato. Then, we performed immunohistochemistry staining of Calb1 (green) in Calb1-IRES2-Cre-D::Ai9 mice. We found that tdTomato-labeled cells (tdTomato⁺ cells) were concentrated in the superficial layer of vCA1 PCs which was consistent with the results of *in situ* hybridization of CALB1 in Allen Brain Atlas (<http://mouse.brain-map.org/experiment/show/71717640>). By immunohistochemistry staining of Calb1, we found that tdTomato⁺ cells (red) colocalized with most Calb1⁺ neurons (green) especially in the superficial layer of vCA1 PCs, indicating efficient labeling *via* Cre/lox strategy in our Calb1-Cre line (sFigure 4).

We also detected a sparse population of tdTomato⁺/Calb1⁻ cells in vCA1. This might

partially attribute to the difference in labeling sensitivity and efficiency between immunohistochemistry staining and genetical labeling *via* Cre/lox strategy. Based on the acceptable recombination specificity and sensitivity, we employed Calb1–Cre line for vCA1^{Calb1+} manipulation experiments in the present study.

Supplementary Figure 4. Characterization of Calb1–IRES2–Cre–D::Ai9 mice.

Example images showing the genetically–tdTomato labeled cells (red) stained with monoclonal antibody against Calb1 (green) in the vCA1 of Calb1–IRES2–Cre–D::Ai9 mice. Scale bar, 50 μ m.

To further confirm the efficiency of Cre/lox strategy in the GCaMP6–targeted vCA1^{Calb1+} neurons, we injected AAV–DIO–GCaMP6–GFP into the vCA1 of Calb1–IRES2–Cre–D::Ai9 mice and quantified the colocalization rate of GFP and tdTomato. Five weeks later, we found that ~94% GFP was colocalized with tdTomato (sFigure 8), indicating an efficient targeting on the vCA1^{Calb1+} neurons with GCaMP6.

Supplementary Figure 8. Characterization of vCA1^{Calb1}-GCaMP mice.

(a) In Calb1-IRES2-Cre-D::Ai9 mice, AAV-EF1a-DIO-GCaMP6f virus was injected into the vCA1. The representative image of the injection site. Exogenously expressed GCaMP6f (green) was colocalized with Calb1 (tdTomato) in the vCA1. (b) Quantitative analysis of colocalization rate (yellow/green) in the vCA1 of vCA1^{Calb1}-GCaMP mice. Scale bar=100 μ m

• SuppFig8 is confusingly labeled, but is also not a convincing control for the level of initial presence of Calb+ cells and the change they see with stimulation. This approach doesn't control for the change in calbindin expression, which should be the outcome measure rather than the overlap with cFos. Obviously cells will have higher cFos after you stimulate them and you're more likely to see overlaps with anything else you stain for, but if they see the same increase in the actual calbindin concentration or can otherwise show a change in plasticity in the circuit that would be more persuasive.

[Response]: We apologize for the misleading. To address the reviewer's concern, we performed Western blot to measure the protein level of Calb1 in vCA1. Quantitative analysis data showed there is no difference among groups (sFigure 13c, d). These data suggest that opto-activation of pBLA-vCA1 inputs did not significantly change the expression of Calb1 protein in the vCA1 neurons. Furthermore, we conducted co-labeling of Calb1 with c-Fos in an independent set of mice 90 min after EPM test. To improve the quality of the images, we chose another Calb1 antibody (ab11426, abcam,

which has been used in over 47 publications as a specific marker for Calb1). Consistent with our previous finding, the number of c-Fos+(green)/Calb1+(red) double-positive neurons (yellow) significantly decreased in the vCA1 of APP/PS1 mice as compared with Wt mice, and photoactivation of pBLA–vCA1 inputs remarkably increased the co-localization of Calb1+ and c-Fos+ neurons in the vCA1 of APP/PS1 mice. With addition of these new data, we feel more comfortable to claim that photostimulation on pBLA–vCA1 circuit can ameliorate the insufficient activation of vCA1^{Calb1+} neurons in APP/PS1 mice.

Supplementary Figure 13. Optogenetic stimulation ameliorates the inhibition of pBLA–vCA1 circuit without changing Calb1 level in APP/PS1 mice.

(a) The representative co-staining of Calb1+/c-Fos+ neurons in the vCA1 of wildtype (Wt) and APP/PS1 mice after photoactivation of pBLA–vCA1 terminals. Scale bar, 50 μm. (b) The number of Calb1+/c-Fos+ neurons in the vCA1 of APP/PS1 mice significantly decreased compared with Wt mice, and photoactivation of pBLA–vCA1 inputs remarkably increased co-localization of Calb1+ (red) and c-Fos+ (green) neurons in the vCA1 of APP/PS1 (one-way ANOVA, $F_{(2, 27)} = 33.77$, $P < 0.0001$, Tukey's multiple comparisons test, $*P < 0.05$ vs Wt, $## P < 0.01$ vs OFF). (c) Western blotting to measure the protein level of Calb1. (d) Quantitative analysis of the protein level of Calb1 in the vCA1.

- Lastly, while previous papers used different methods to look at BLA-vHPC projectors and also the general activity of vHPC CA1 cells in approach/avoidance and other tasks, they have not reported the type of clear spatial segregation of encoding along the superficial-deep axis that the optogenetic and calcium imaging results in this paper suggest would be present. This again needs to be directly addressed.

[Response]: We agree with the reviewer that to perform the population calcium recording and optogenetic manipulation on vCA1^{Calb1-} neurons as we did on vCA1^{Calb1+} neurons will be a significant step forward in uncovering the type of spatial segregation of encoding along the superficial–deep axis of vCA1. In the present study, we did perform population calcium recording combined with retrograde tracing. We found that the aBLA–vCA1^{Calb1-} circuit was specifically activated when mice moved from closed arm to open arm at the transition zone (Figure 6a) which was distinct from the firing pattern of pBLA–vCA1^{Calb1+} circuit (Figure 6c) in EPM. Due to the dominant innervation of aBLA on the Calb1⁻ neurons in the deep layer of vCA1 PCs (Figure 1), it is possible that at least part of vCA1 neurons in deep layer contribute to the observed avoidance behavior during conflict environment. As the main focus of the current study is on the pBLA–vCA1^{Calb1+} circuit, we hope the reviewer agrees that the characteristic superficial and deeper layer of vCA1 neurons and the functions of vCA1^{Calb1-} neurons will deserve a separate investigation.

Reviewer #2 (Remarks to the Author):

The manuscript tests the hypothesis that BLA projections to the ventral HPC are composed of distinct pathways - an anterior input to the deep HPC and a posterior input to the superficial HPC. The authors provide substantial evidence that these two projections form unique roles in anxiety-like behavior in rodents. In addition, the authors propose a novel hypothesis regarding Alzheimer's disease and anxiety-like behavior reported in rodent APP models. The manuscript is well written, however several methodological details are missing from the manuscript.

Response: We sincerely appreciate the reviewer's positive comments on our manuscript.

My major comments are listed below.

1. It is a bit difficult to determine which region of vCA1 is included in Figure 1. Could the authors provide a low magnification image and with a box indicating where the higher magnification image was taken? Also, did the projections extend into intermediate HPC?

[Response]: We thank the reviewer for the comment. As suggested, we have now provided the uncropped versions of low magnification images (please see sFigure 1). Using H129 anterograde tracing system, we detected robust GFP signals in the ventral hippocampal CA1 (sFigure 1c, g). Interestingly, we did not observe obvious tracing signals in the dorsal (sFigure 1a, e) and intermediate (sFigure 1b, f) hippocampus at 24 h after H129-G4 injection in aBLA or pBLA. These histological evidence suggest that a/pBLA may predominantly innervate ventral hippocampus as compared with the dorsal and intermediate hippocampus.

Figure 1. Trans-synaptic virus tracing a/pBLA-vCA1 circuits. (a-g) Anterograde

H129–G4 was injected into the aBLA (**a**) and pBLA (**f**), respectively. 24 h later, mice were sacrificed for series slicing. Representative images of aBLA–(**a–c**) and pBLA–projecting (**e–g**) regions by serial coronal sections. Scale bar=1 mm. (**d, h**) Higher–resolution images of the boxed regions in panel **c** and **g**, showing that H129–G4 injected in aBLA preferentially label the deep layer of vCA1 PCs (**d**), while in pBLA predominantly label the superficial layer of vCA1 PCs (**h**). Scale bar=100 μ m. (**i–l**) Comparison of the projection strength of aBLA and pBLA to vCA1: (**i**) Quantitative analysis of the starter cells from anterior (A) to posterior (P) axis in aBLA–H129 and pBLA–H129 mice, respectively. Note that the maximum values of the two groups (aBLA–H129 group at bregma -1.46 vs pBLA–H129 group at bregma -2.46) was identical. $n=7$ per group, unpaired t test, $t=1.842$ $df=11.99$, $P=0.0903$. (**j**) Comparison of the number of H129–labeled vCA1 cells in aBLA–H129 and pBLA–H129 mice, showing that more vCA1 neurons were innervated by pBLA than aBLA from bregma -3.08 to -3.64. $n=7$ per group, two–way ANOVA, $F(6, 72) = 5.103$, $P=0.0002$, Sidak's multiple comparisons test, $*P < 0.05$, $**P < 0.01$ vs pBLA–H129. (**k, l**) Comparison of the number of deep (**k**) and superficial (**l**) vCA1 cells labeled by H129 in aBLA–H129 and pBLA–H129 mice. From bregma -2.92 to -3.64, more H129–labeled cells were located in the deep layer of vCA1 PCs in aBLA–H129 mice as compared with those in pBLA–H129 mice (**k**). While the number of H129–labeled superficial vCA1 cells was greater in pBLA–H129 mice than those in aBLA–H129 mice (**l**). $n=7$ per group, Two–way ANOVA, deep layer of vCA1 PCs: $F(6, 72) = 6.96$, $P<0.0001$; superficial layer of vCA1 PCs: $F(6, 72) = 22.41$, $P<0.0001$, Sidak's multiple comparisons test, $**P < 0.01$ vs pBLA–H129 (**k**) or aBLA–H129 (**l**). so, stratum oriens; sp, stratum pyramidale; sr, stratum radiatum.

2. Figure 1 data - A definition for the superficial-deep axis of vCA1 should be provided in the Methods section or in the results section where it is first discussed. This could also be illustrated in a schematic and presented early in the results section or perhaps as a panel in Figure 1.

[Response]: We thank the reviewer for suggesting the details. We have now clarified that superficial layer of CA1 pyramidal cells (PCs) is a subdivision adjacent to the stratum radiatum, while the deep layer of CA1 PCs is adjacent to stratum oriens in the result section (please see page 6 line106 and 109). We also provided a schematic to illustrate superficial and deep layer of vCA1 PCs in sFigure 1(d, h) and Figure 1(panel a, e, f). Indeed, the supplementary description and schematic have greatly improved the intelligibility of the paper.

3. OFT - The approximate size of the central zone should be described.

[Response]: We apologize for the insufficient description. In the methods section, we have clarified that the open field chamber was made of transparent plastic (50 cm × 50 cm × 40 cm, length × width × height) and divided into a central field (25 cm × 25 cm) and a periphery field (please see page 23 line 557 to 560). Thus, the central zone area was defined as 50% of the open field arena.

4. In Figure 4, Ca imaging data is presented showing an increase in overall fluorescence when entry into the closed arm is available. On the other hand, overall fluorescence decreases when entry into the open arm is made available. The relationship between population Ca fluorescence is only correlated with time (as shown in Fig. 4). Does time represent the animals entry into the open/closed arms or does it represent when the doors to the arms were opened? Was motion tracking used to monitor the position of the animal as it entered the arms - was this done manually with video records or using tracking software? It is unclear how the neural activity was correlated with the animals behavior.

[Response]: We appreciate for the opportunity to clarify this point. In our study, the time point at which mouse entered the closed or open arm was set as 0. Photometry

signals were exported as average peri-event time histograms around the events, and specific behavioral periods, *i.e.*, baseline (a 5 s period beginning 15 s before entry), pre-enter (a 5 s interval from beginning 8s before entry) and enter (a 1s after entry), were extracted for analysis (please see methods section, page 24, line 603 to 607). Motion tracking and manual tagging were used to monitor and mark the position of the animals when they were exploring in the EPM.

We agree with the reviewer that exporting photometry signals correlated with animals' position will help to decode the information carried out by specific neurons in EPM. However, the main focus of our current study was to explore whether and how pBLA-vCA1^{Calb1+} connection could control avoidance-approach behaviors during decision-making processes. Thus, we set a short observation time window around transition zone, *i.e.*, before (15 s) and after (1 s) the decision-made (Figure 4b-g), to record the activity of pBLA-vCA1^{Calb1+} connection. To maximally link the neural activities with assessment during decision, we used the rolling average calculation to identify the "rolling" picture of population calcium signals over time as mice started and finished decision-making around transition zone in EPM. We found that vCA1^{Calb1+} was significantly activated when mice entered closed arm (Figure 4b-d), while decreased as soon as mice entered open arm (Figure 4e-g). These data indicate a close relationship between the activities of vCA1^{Calb1+} neurons and approach-avoidance behavior in conflict situation. Then, we photoactivated vCA1^{Calb1+} neurons at transition zone and found a significant promotion on approach behavior in EPM (Figure 4j, k). Based on the data of population calcium recording and optogenetic manipulation, we can conclude that the activities of vCA1^{Calb1+} neurons control approach behavior in conflict exploratory task. Thus, we exported the population calcium fluorescence correlated with time not merely with the location of mice in open arm/closed arm in population calcium recording experiment. We have now extended the result to address this point (page 9, line 193 and 196; page 24, line 603 to 609).

5. Some additional details regarding the relationship between Ca transients and the animals behavior is warranted. Did the authors examine the locomotor speed of the animal when they transitioned between open/closed arms? Perhaps differences in locomotor variables are driving this population response as shown in Fig 4 and 6. Further, it is possible that locomotor differences between APP and WT animals are driving the different Ca populations responses shown in Fig. 6.

[Response]: Thanks to the reviewer for bring up this very interesting issue. We agree with the reviewer that locomotor differences may induce variables in the calcium response. To address this concern, we examined the velocity of the animals when they transitioned between open/closed arms (please see sFigure 12). The results showed that the velocity of movements maintained at a low level in the baseline and pre-enter period, but significantly increased in the enter period as the wild-type mice transitioned from the closed to open arm (sFigure 12c,f). These relatively low moving speed in the baseline and pre-enter period may indicate risk assessment behavior and decision-making dithering between approach and avoidance under conflict situation. Thus, we think that the movement-related calcium activity observed around transition zone in EPM may attribute to the approach-avoidance decision-making controlled by a/pBLA neurons in the a/pBLA-vCA1 circuits. In the AD mice, the speed changing pattern around transition zone was identical to the wild-type mice (sFigure 12c,f). These supplementary data indicate that the disorganized calcium transients in APP/PS1 mice were not caused by their locomotor variables, but were closely related with the avoidance-approach imbalance in the AD mice. We have now added the results of velocity as supplementary Figure 12 and clarified in results section (please see page 13 line 300 to 307).

Supplemental Figure 12. Velocity of movements during photometry experiments in mice.

(a–d) Velocity of Wt **(a, d)** and APP/PS1 **(b, e)** mice from the closed arm to open arm in EPM. **(c, f)** Average velocity during baseline, pre–enter, and enter periods from **a, b** and **d, e**. Two–way RM ANOVA for Wt and APP/PS1 respectively, ** $P < 0.01$ vs baseline; ## $P < 0.01$ vs pre–enter **(c, f)**. Two–way RM ANOVA for Wt vs APP/PS1, aBLA: $F_{(2, 36)} = 2.388, P = 0.1062$; pBLA: $F_{(2, 40)} = 3.059, P = 0.0581$. Data are presented as mean \pm SEM.

Reviewer #3 (Remarks to the Author):

The authors report that projections from the anterior and posterior basolateral amygdala (aBLA and pBLA, respectively) to the ventral hippocampus CA1 (vCA1) have opposite functions. Activation of pBLA-vCA1 and aBLA-vCA1 respectively decrease and increase anxiety in mice. Furthermore, the authors show that pBLA targets calb1+ neurons, while aBLA targets calb1- neurons. Photometry recordings show that calb1+ vCA1 neurons are activated when mice enter the safe closed arms of the maze in normal mice, but not in a model of Alzheimer's Disorder (AD). AD mice also show distinct transcriptional changes in aBLA and pBLA compared to control mice. Lastly, AD mice show increase anxiety and lower performance in the Barnes Maze. Both of these deficits can be rescued by optogenetic activation of pBLA-vCA1 in control mice, but not in mice that show reduced levels of calb1 in vCA1.

The main novelty in this paper is showing that aBLA and pBLA projections to the hippocampus activate non-overlapping and molecularly distinct (calb1+ and calb1- cells) populations, and that these projections influence anxiety in opposing ways. These findings are important and of interest to the general systems neuroscience community. However, there are several points that must be addressed prior to publication.

[Response]: We greatly appreciate the reviewer's positive comments on our manuscript.

1. The authors often use incorrect and awkward phrasing. Sometimes this problem obscures what the authors want to convey. There are dozens of occurrences of grammatical errors, such as:

“inputs with disordered firings and heterogenized protein networks”

“the decreased approaching exploration renders them failed to extinct the previous fear’

“ventral hippocampal CA1 (vCA1) is identified as anxiety cell niche”

This article cannot be published unless it the writing is thoroughly corrected by a

neuroscientist that can write well in English.

[Response]: We apologize for the poor writing. As suggested, we have carefully revised the paper (highlighted in red) and have asked professional editing services for proofreading of the manuscript. We believe that the clarity of the paper has been substantially improved.

Specifically, the above sentences have been corrected as follows:

changed “inputs with disordered firings and heterogenized protein networks” into “..., insufficient activation of pBLA–vCA1 inputs with disorganized firings and abnormal protein networks in a/pBLA were shown, ...” (please see page 3 line 37 and 38);

changed “the decreased approaching exploration renders them failed to extinct the previous fear’ into “the decreased approaching exploration fails to endow them with the capacity to extinct the previous fear” (please see page 4 line 53 and 54);

changed “ventral hippocampal CA1 (vCA1) is identified as anxiety cell niche” into “anxiety cells were discovered in ventral hippocampal CA1 (vCA1)” (please see page 4 line 66 and 67).

2. Calbindin expressing cells in the hippocampus can be found both amongst pyramidal cells (Kohara et al., 2014), as well as GABAergic interneurons in CA1 (Jinno and Kosaka, 2006). The authors should show if the calb1+ cells that receive pBLA inputs are gabaergic or glutamatergic. If they are gabaergic, the authors could show if these cells inhibit the calb1- cells that receive aBLA input. These data would provide a very nice mechanistic explanation for the opposing functions of the aBLA and pBLA input to vCA1.

[Response]: Thanks to the reviewer for the constructive suggestions. As suggested, we have performed GAD67 (GABAergic interneuron marker) staining combined with anterograde tracing. The results showed that H129–labeled vCA1 neurons (green) were barely colocalized with GAD67, indicating that the GABAergic interneurons in vCA1 were not predominately innervated by pBLA. Considering that immunohistochemical staining cannot completely exclude the possibility of local inhibition, we have made great efforts to measure the functional interaction between aBLA–innervated and

pBLA-innervated vCA1 neurons *via* electrophysiological recording in combination with anterograde tracing. However, we found that it is extremely hard to patch aBLA-innervated and pBLA-innervated vCA1 neurons simultaneously after H129 anterograde tracing. This may attribute to the unhealthy condition of the labeled neurons in HSV-antetrograde tracing system and/or experimental difficulties of double patch. By literature searching, we have been unable to find any previous reports shown electrophysiological recording data on H129 infected neurons, to date. Therefore, we are currently unable to provide further direct evidence to identify the reviewer's speculation due to the technique limitation. We deeply appreciate the reviewer's insightful suggestions and believe that this speculation deserves investigation in the future as the technology develops. Thus, we added this speculation in the discussion to extend our understanding about the possible mechanisms (please see page 17 line 420 to 424). We think that in addition to the possibility that pBLA-innervated vCA1 locally inhibits aBLA-innervated vCA1 neurons, the distinct downstreams of vCA1^{calb1+} and vCA1^{calb1-} cells can also contribute to their opposite function in anxiety. We have discussed this in the revised paper (please see page 17 line 415 to 424).

The pBLA barely innervates GABAergic interneurons in the vCA1. Antegrade H129-G4 was injected in the pBLA to label pBLA-innervated vCA1 neurons. One day later, pBLA-H129 mice were sacrificed and stained with GAD 67 (GABAergic interneuron marker). GFP (green) were barely colocalized with GAD67 (red), indicating that pBLA barely innervates GABAergic interneurons in the vCA1.

3. The authors repeatedly state that calb1+ vCA1 cells “encode “secure” information”. This statement implies that these cells are constantly activated during exposure to safe locations. However, their own data in Fig. 4b and 4e shows that these cells are activated during the entry to the closed arm, not during the whole time the mouse is in the closed arm. The authors should change the text throughout the manuscript and state instead that these cells are activated immediately prior to entry into the closed arm.

[Response]: Thanks to the reviewer for pointing out this important issue. We agree with the reviewer that vCA1^{Calb1+} neurons should be constantly activated during exposure to safe locations if they encoded “secure information”. In the present study, we set a short observation time window, *i.e.*, before (15 s) and after (1 s) decision made (please see methods, page 24 lines 603-607 and Figure 4b–g) to record the activity of pBLA–vCA1^{Calb1+} connection. Our data suggest a close correlation between the activity of vCA1^{Calb1+} neurons and approach–avoidance decision. Since the average calcium activity (5 s period around transition point) in the closed arm was much greater than that in the open arm (Figure 4 h), we speculate that the approach–avoidance decision may depend on the assessment of aversion/safe information. Together with our optogenetic manipulation data, we deleted all the expressions of “encode secure information” and changed them into “the assessment of aversion/safe information in conflict situation” (please see page 10 line 208 to 211).

4. I haven't checked the references throughout the paper, but I found a few incorrect citations. For example: “theta–frequency firing within ventral hippocampus (vHPC) is synchronized with medial prefrontal cortex (mPFC) discharge during exposure to the anxiogenic environments^{2,14}”

References 2 and 14 do not have any data related to theta-frequency firing. Please carefully check if all citations are correct.

[Response]: We deeply appreciate the reviewer's comment and apologize for the inappropriate citations. We have now replaced the citation 2 and 14 by “Adhikari A, Topiwala MA, Gordon JA. Synchronized activity between the ventral hippocampus and the medial prefrontal cortex during anxiety. *Neuron* 65, 257-269 (2010).” (please see

page 4 line 63 to 65). We have also carefully checked all the reference papers and have maximally killed the inappropriate citations.

5. The authors state that “L-type maze has definite moving direction which can help the investigators to accurately determine the animals’ decision-making behaviors.” It is not clear from this sentence why the authors use the L-maze instead of the more common plus maze. Please clarify.

[Response]: We apologize for the insufficient description. The main purpose of optogenetic manipulation in $vCA1^{Calb1+}$ mice was to investigate whether activation of $vCA1^{Calb1+}$ neurons drove approach behaviors to the open arm at the transition zone. To reach this goal, a simple paradigm which is able to clearly identify the avoidance- or approach-decision-making in the transition zone is important. The “L” maze invented by the current study consists of one open arm and one closed arm, therefore, there are only two choices when mouse leaves the transition zone, *i.e.*, going to the opposite arm or going back to the previous arm. On the other hand, the classical plus maze provides multiple choices, *i.e.*, going back, going to the opposite arm, turning left, turning right and transitory switch among these directions, which can prolong the decision latency of the mice. More importantly, hesitating behaviors of the mouse around the transition zone can make the experimenter too confused to accurately determine and interpret the animals’ decision. As suggested by the reviewer, we have now clarified the advantage of “L” maze over plus maze in the revised paper (please see page 23, line 565 to 572).

6. The authors wrote that “stimulation of pBLA-vCA1 inputs in APP/PS1 mice may break the vicious cycle to ameliorate AD anxiety by promoting approach to regain the avoidance-approach balance”. This statement implies that temporary stimulation of pBLA-vCA1 inputs would have some long-term effect to increase approach, but the authors have not done any experiments to show this. The data in Fig 2 actually show that after stimulation is turned off the animals resume normal avoidance, without any long-term stimulation effect.

[Response]: We apologize for the inappropriate interpretation of the results. We agree

with the reviewer that our current data can only indicate a temporary anxiolytic effect of pBLA–vCA1 photoactivation in APP/PS1 mice. Therefore, we changed the sentence into “during stimulation epoch, pBLA–vCA1 inputs can resume normal avoidance in APP/PS1 mice with the mechanisms involving promoting approach to regain the avoidance–approach balance” in the revised paper (please see page 18 line 437 to 439).

7. The authors should write a short sentence describing why they chose APP/PS1 mice as an AD model.

[Response]: We thank the reviewer for the comments. As suggested, we have added the following description in the section of main text to clarify the reason for using APP/PS1 mice in the present study (Please see page 11 line 241 to 245): “A previous study showed that amyloid–beta ($A\beta$) accumulation in the BLA enhances anxiety behavior in AD transgenic mice. To explore whether and how aBLA or pBLA is involved in AD–related anxiety, we first examined $A\beta$ pathologies in the aBLA and pBLA of APP/PS1 mice which carry APP and PS1 mutated genes with overproduction of $A\beta$ in the brain.”.

8. It is unclear how the authors are defining pBLA. For example, Fig 2a and 2g both show pBLA, but the photographs show sections that are in very different positions in the anterior-posterior axis.

[Response]: We thank the reviewer for pointing out this issue. The pBLA is defined as posterior part of BLA. According to the mouse brain atlas, pBLA emerges at bregma -1.94 mm and disappears after bregma -3.16 mm (please see the Figure below). At bregma -2.3 mm (as showed in Figure 2a, g), the left (Figure 2g) and right (Figure 2a) pBLA are quite different from each other in their shapes. To illustrate our representative image more accurately, we have added “right” and “left” on panel a and panel g respectively in Figure 2. Furthermore, we emphasized our counterbalance manipulation on the left and right pBLA in the methods section to clarify the targeting manipulation (please see page 22 line 543).

[REDACTED]

Coordinates of pBLA in mouse' brain. (a) pBLA emerges at bregma -1.94 mm and disappears after bregma -3.16 mm. (b) Left and right pBLA are quite different from each other in their shapes at bregma -2.3mm. (c) ChR2 and NpHR were respectively expressed in the pBLA as shown in Figure 2 panel g and a.

9. In Fig 1a the authors use viruses to perform anterograde monosynaptic tracing. While this method has been used before, it has not been used extensively and validated by many researchers. Please show in acute slice experiments that anterogradely tagged cells respond to optogenetic stimulation of BLA inputs while non-tagged cells in the same layer do not respond to stimulation of BLA inputs.

[Response]: Thanks to the reviewer for the constructive suggestion. We agree with the reviewer that showing the response of non-tagged cells in the same layer upon optogenetic activation of a/pBLA inputs is a perfect control for tagged cells in a/pBLA–vCA1 circuits. It is also helpful to precisely dissect the functional innervations from a/pBLA to vCA1 in the same layer of vCA1. However, one limitation of the monosynaptic tracing system in this field is that TK complementation leads to severe cell damage and cytopathy of starter cells in the injection site⁶. Thus, we were unable to activate the unhealthy axons of a/pBLA starter cells to elicit responses in vCA1 neurons after anterograde tracing.

Nonetheless, we added a negative control in the electrophysiology experiments in

Calb1-IRES2-Cre-D::Ai9 mice (fluorescently label Calb1⁺ neurons with tdTomato) to at least partially address the reviewer's concern. It was shown that in the superficial layer of vCA1 PC, vCA1^{tdTomato+} did, while vCA1^{tdTomato-} neurons did not respond to the photoactivation of pBLA-vCA1 terminals in the vCA1 (sFigure 6). These data support the findings that pBLA preferentially innervated Calb1⁺ neurons in the vCA1. These new data have substantially strengthened our previous findings.

Supplementary Figure 6. Excitatory pBLA neurons innervate vCA1^{Calb1+} not vCA1^{Calb1-} neurons. AAV5-CaMKIIa-hChR2(H134R)-EYFP was injected into pBLA of Calb1-IRES2-Cre-D::Ai9 mice. Four weeks later, electrophysiological response was recorded from vCA1^{Calb1+} and vCA1^{Calb1-} neurons respectively upon the photoactivation of pBLA-vCA1 inputs. (a) Representative images of patch pipette tip on a vCA1^{Calb1+} (tdTomato⁺) pyramidal neuron in a hippocampal slice (scale bar=30 μ m). (b) Representative EPSCs, evoked by light stimulation of the pBLA-vCA1 inputs, were recorded from Calb1⁺ and Calb1⁻ neurons in vCA1.

Reference

1. Kim J, Zhang X, Muralidhar S, LeBlanc SA, Tonegawa S. Basolateral to Central Amygdala Neural Circuits for Appetitive Behaviors. *Neuron* **93**, 1464-1479 e1465 (2017).
2. Beyeler A, *et al.* Divergent Routing of Positive and Negative Information from the Amygdala during Memory Retrieval. *Neuron* **90**, 348-361 (2016).
3. Tye KM, *et al.* Amygdala circuitry mediating reversible and bidirectional control of anxiety. *Nature* **471**, 358-362 (2011).
4. Sadrian B, Wilson DA. Optogenetic Stimulation of Lateral Amygdala Input to Posterior Piriform Cortex Modulates Single-Unit and Ensemble Odor Processing. *Frontiers in neural circuits* **9**, 81 (2015).
5. McDonald AJ, Zaric V. GABAergic somatostatin-immunoreactive neurons in the amygdala project to the entorhinal cortex. *Neuroscience* **290**, 227-242 (2015).
6. Zeng WB, *et al.* Anterograde monosynaptic transneuronal tracers derived from herpes simplex virus 1 strain H129. *Molecular neurodegeneration* **12**, 38 (2017).
7. Felix-Ortiz AC, Beyeler A, Seo C, Leppla CA, Wildes CP, Tye KM. BLA to vHPC inputs modulate anxiety-related behaviors. *Neuron* **79**, 658-664 (2013).
8. Kim SY, *et al.* Diverging neural pathways assemble a behavioural state from separable features in anxiety. *Nature* **496**, 219-223 (2013).
9. Padilla-Coreano N, *et al.* Direct Ventral Hippocampal-Prefrontal Input Is Required for Anxiety-Related Neural Activity and Behavior. *Neuron* **89**, 857-866 (2016).
10. Felix-Ortiz AC, Burgos-Robles A, Bhagat ND, Leppla CA, Tye KM. Bidirectional modulation of anxiety-related and social behaviors by amygdala projections to the medial prefrontal cortex. *Neuroscience* **321**, 197-209 (2016).

Reviewers' comments:

Reviewer #1 (Remarks to the Author):

- Summary: This paper seeks to elucidate a potentially important heterogeneity in the BLA-vHPC circuit related to approach/avoidance behavior building off previous work from their lab and others. Their novel finding that anterior and posterior BLA-vHPC projections differentially modulate behavior in the EPM and other tests is an interesting, potentially clinically relevant addition to the growing literature on how related circuitry might be manipulating these behaviors. However, their central finding that pBLA and aBLA-vHPC projectors are separate populations with opposite effects on approach/avoidance behavior is at odds with multiple papers they cite in recent literature. That is not a problem, as revision of the literature is always welcome, however it is necessary to substantiate these claims, and correctly interpret the previous results. The most important thing here is the necessity to provide sufficient histological/electrophysiological evidence to demonstrate the specificity of injections and the behavior

- o This paper needs to be edited for clarity. This was brought up in the first round of review. I appreciate the authors have sent to a proofreader, but this wasn't sufficient. There are numerous grammatical errors and confusing or incorrect sentences, making this paper incredibly difficult to read/assess.

- o Need to show a detailed accounting of the injection spread for their behavioral experiments, specifically in the AP spread of BLA (and other subfields) as well as the spread of the projections in vHPC. A detailed description of the injection sites for the opto experiments in figures 2 and 3 is absolutely essential for interpreting these results. In this revision, they have provided one image of the injection site for each experiment, but no sense about the spread into the other portion of the amygdala. As the primary point of this paper is that these two portions of BLA transmit different signals, it is absolutely crucial to know what cells they actually infected in each experiment. (They authors may have misunderstood this point in the first round of review, and thus have provided their HSV infection, but this is a different experiment)

- o Again, as brought up in the first round of review, the specificity of the monosynaptic anterograde tracing is not provided. If the HSV they use here has any retrograde activity, as this is a reciprocal circuit, this would invalidate this technique. The slice electrophysiology is a much better way to assess this, but they only show one example cell in Fig S6, so I would suggest the authors greatly increase their N for fig S6 so there is a conclusion on the percentage of Calb+/- cells that get input from one area or the other.

- o A number of the papers they cite either broadly targeted BLA or vHPC and see the same optogenetic effects as they claim to for the aBLA projectors. Again given that multiple previous papers have claimed that aBLA is a far less significant population than the pBLA, it seems more likely that the previous results would have been primarily inhibiting the pBLA projectors, and that is what is suggested by the histology shown in some of those papers, particularly the Felix-Ortiz et al. 2013 paper. This should be discussed.

- o The authors repeatedly generalize their findings in troubling ways, most of which might be due to their issues with the losing clarity in the writing, but needs to be properly addressed in the edits.

- ♣ For example, "Although negative and positive emotion were respectively elicited after aBLA and pBLA manipulation..."

- ♣ Again, there is nothing in the data that supports their claim that the two projections differ in their strength (their claim in figure 7).

- ♣ Reference #1 seems like a strange choice when making a broad claim about emotion and decision making.

- This was brought up in the first round, It's not clear that the second half of the paper adds much to the narrative and is a bit hard to interpret. To make sense of the shRNA manipulation before doing this test in the APP/PS1 mice they should've done a control in their WT cohorts to see if the shRNA injection has the same effect on the optogenetic anxiogenic and anxiolytic effects.

- SFig13 doesn't seem to add much to the study. Of course stimulation is going to increase the number of cells expressing Fos, not clear what this means for the APP/PS1 phenotype

Reviewer #2 (Remarks to the Author):

The authors have addressed my comments and suggestions for improvement.

Reviewer #3 (Remarks to the Author):

Overall the authors have done a good job at addressing the reviewer's concerns. Only a few problems remain unsolved. Despite these small problems, the results in general are interesting and I believe the manuscript can be accepted as it is.

1. The writing is better, but the quality is still considerably below average for this journal. In the initial submission the writing quality sometimes precluded understanding of what the authors wanted to communicate. Now it is possible to understand everything, but there are still many instances of awkward and incorrect phrasing (such as "disorderly firing"). For example, the labels in Fig 5 are very strange. What is "single organism process"? Fig 7a should use "Structural heterogeneity" and "functional heterogeneity" instead of "structure" and "function heterogeneity".

2. Fig 1i and 1j. Please add data showing the responses are blocked after adding TTX (TTX alone, in the absence of 4-AP).

Please find below a point-by-point reply (in black) to the reviewers' comments (in blue):

Reviewer #1 (Remarks to the Author):

• Summary: This paper seeks to elucidate a potentially important heterogeneity in the BLA-vHPC circuit related to approach/avoidance behavior building off previous work from their lab and others. Their novel finding that anterior and posterior BLA-vHPC projections differentially modulate behavior in the EPM and other tests is an interesting, potentially clinically relevant addition to the growing literature on how related circuitry might be manipulating these behaviors. However, their central finding that pBLA and aBLA-vHPC projectors are separate populations with opposite effects on approach/avoidance behavior is at odds with multiple papers they cite in recent literature. That is not a problem, as revision of the literature is always welcome, however it is necessary to substantiate these claims, and correctly interpret the previous results. The most important thing here is the necessity to provide sufficient histological/electrophysiological evidence to demonstrate the specificity of injections and the behavior.

[Response]: We thank the reviewer for the positive evaluation of our manuscript. As suggested, we performed additional experiments and revised the manuscript accordingly to address the concerns.

Issues:

1. This paper needs to be edited for clarity. This was brought up in the first round of review. I appreciate the authors have sent to a proofreader, but this wasn't sufficient. There are numerous grammatical errors and confusing or incorrect sentences, making this paper incredibly difficult to read/assess.

[Response]: We thank the reviewer for the suggestion. To improve the writing, we have asked editing service (Springer Nature) to make further professional editing of our manuscript (highlighted in red). Now, we believe that the clarity of the paper has been substantially improved.

2. Need to show a detailed accounting of the injection spread for their behavioral experiments, specifically in the AP spread of BLA (and other subfields) as well as the spread of the projections in vHPC. A detailed description of the injection sites for the opto experiments in figures 2 and 3 is absolutely essential for interpreting these results. In this revision, they have provided one image of the injection site for each experiment, but no sense about the spread into the other portion of the amygdala. As the primary point of this paper is that these two portions of BLA transmit different signals, it is absolutely crucial to know what cells they actually infected in each experiment. (They authors may have misunderstood this point in the first round of review, and thus have provided their HSV infection, but this is a different experiment)

[Response]: We thank the reviewer for the constructive comments and detailed suggestions. Now, we have evaluated the injection spread in the brain (sFigure 7 and sFigure 9). Again, robust eYFP expression was detected in the pBLA (sFigure 7d, e) and aBLA (sFigure 9b, c) subregions after their injection, respectively, which provides evidence for the precise targeting on pBLA and aBLA in our experiments. Axonal fibers originating from Chr2-eYFP-expressing projection neurons in the pBLA and aBLA were identified as green fluorescence in coronal brain sections (sFigure 7, sFigure 9). In aBLA tracing, the projection signals were predominantly detected in the Au1, AuV, TeA, Ect, PRh, LEnt, APir, PP, SNL, PIL and vCA1 (sFigure 9e-g, i). In line with the findings from anterograde tracing (Figure 1), the intensity of Chr2-eYFP-expressing fibers in the deep layer were much stronger than those in the superficial layer of the vCA1 (sFigure 9h). However, in pBLA tracing, the projection signals were obviously detected in LEnt, APir, vCA1, Ect, TeA, and PMCo (sFigure 7f, g, i). Unexpectedly, there was no difference in projecting density between the superficial and deep layer of the vCA1 (sFigure 7h). We speculate that the fibers observed in the deep layer may not innervate the neurons there, but pass through the deep layer to make connections with the vCA1 neurons in the superficial layer. This speculation was further supported by evidence from slice electrophysiology (sFigure 6). We believe that our new results from a detailed accounting of the injection spread of BLA along its A-P axis and the spread of the projections in vHPC have provided quantitative description

of the targeting of aBLA–vCA1 and pBLA–vCA1 circuits in Figures 2 and Figure 3, as suggested by the reviewer.

Supplementary Figure 7. Anatomical characterization of projections arising from

pBLA. (a-g) Representative image showing Chr2-eYFP expression (green) localized to pBLA and the projections arising from the pBLA. **(h, i)** Quantitative analysis of the

fluorescence intensity in pBLA (injection site) and its projection area **(i)**, especially in the superficial and deep layer of the vCA1 **(h)**. n=7 mice per group, paired *t* test,

t=0.4737, df=6, P=0.6524. Data are presented as the mean ± SEM. CPu, caudate

putamen; Pir, piriform cortex; La, lateral amygdaloid nucleus; Au1, primary auditory

cortex; AuD, dorsal area of secondary auditory cortex; AuV, ventral area of secondary

cortex; AuD, dorsal area of secondary auditory cortex; AuV, ventral area of secondary

cortex; AuD, dorsal area of secondary auditory cortex; AuV, ventral area of secondary

auditory cortex; TeA, temporal association cortex; Ect, ectorhinal cortex; PRh, perirhinal cortex; Lent, lateral entorhinal cortex; APir, amygdalopiriform transition area; PMCo, posteromedial cortical amygdaloid nucleus; PIL, posterior intralaminar thalamic nucleus; PP, peripeduncular nucleus; SNL, lateral part of substantia nigra; s, superficial; d, deep. Scale bar, 1 mm or 100 μ m (f, g enlarged box).

Supplementary Figure 9. Anatomical characterization of projections arising from aBLA. (a-g) Representative image showing ChR2-eYFP expression (green) localized

to aBLA and the projections arising from the aBLA. **(i, h)** Quantitative analysis of the fluorescence intensity in aBLA (injection site) and its projection area **(i)**, especially in the superficial and deep layer of the vCA1 **(h)**. $n=7$ mice per group, paired t test, $t=7.261$, $df=6$, $P=0.0003$. Data are presented as the mean \pm SEM. Scale bar, 1 mm or $100\mu\text{m}$ (**e**, **f** enlarged box).

3. Again, as brought up in the first round of review, the specificity of the monosynaptic anterograde tracing is not provided. If the HSV they use here has any retrograde activity, as this is a reciprocal circuit, this would invalidate this technique. The slice electrophysiology is a much better way to assess this, but they only show one example cell in Fig S6, so I would suggest the authors greatly increase their N for fig S6 so there is a conclusion on the percentage of Calb \pm cells that get input from one area or the other.

[Response]: We thank the reviewer for the comment and suggestions. Now, we performed brain slice recording and made quantitative analysis to address this issue. In the superficial layer of vCA1 PCs, $\sim 77\%$ of tdTomato $^+$ (Calb $^+$) neurons responded to the photoactivation of pBLA–vCA1 terminals (sFigure 6a, b). The EPSCs in the pBLA–vCA1 $^{\text{Calb}^+}$ pathway could be rescued by the application of 4-aminopyridine (4-AP), a blocker of voltage-gated K $^+$ channels, after they were blocked by tetrodotoxin, indicating their monosynaptic origin (sFigure 6b, c; Petreanu et al., 2009). Interestingly, most of tdTomato $^-$ (Calb $^-$) neurons ($\sim 92\%$) kept silent to the photoactivation of pBLA–vCA1 inputs, indicating non-functional connections between pBLA and vCA1 $^{\text{Calb}^-}$ neurons (sFigure 6d, e). Unlike vCA1 $^{\text{Calb}^+}$ neurons, $\sim 81\%$ of Calb $^-$ neurons in the deep layer of vCA1 PCs received functional monosynaptic innervations from aBLA (sFigure 6f-j), suggesting an overwhelming superiority of aBLA–vCA1 $^{\text{Calb}^-}$ connection in aBLA–vCA1 pathway. With addition of these new data, we feel more comfortable to draw the conclusion of monosynaptic aBLA–vCA1 $^{\text{Calb}^-}$ and pBLA–vCA1 $^{\text{Calb}^+}$ pathways. We also extended the results part of this supplementary data on

Supplementary Figure 6. Identification of monosynaptic pBLA-vCA1^{Calb1+} and aBLA-vCA1^{Calb1-} connections. In Calb1-IRES2-Cre-D::Ai9 mice, AAV5-CaMKIIa-hChR2(H134R)-EYFP was injected into the aBLA and pBLA respectively. Four weeks later, electrophysiological response was recorded from vCA1^{Calb1+} and vCA1^{Calb1-} neurons upon the photoactivation of pBLA-vCA1 inputs or aBLA-vCA1 inputs. **(a, f)** Representative images of patch pipette tips on vCA1^{Calb1+} (tdTomato+) and vCA1^{Calb1-} (tdTomato-) pyramidal neurons in hippocampal slices (scale bar=30 μ m). **(b, d, g, i)** Representative traces of EPSCs in the pBLA-vCA1^{Calb1+} pathway **(b, d)** and aBLA-vCA1^{Calb1-} pathway **(g, i)** recorded under different experimental conditions. EPSCs were evoked by photostimulation of Chr2-expressing axons from the pBLA and aBLA, and recorded in Calb1⁺ and Calb1⁻ neurons respectively in the vCA1. Optogenetically-induced and tetrodotoxin (TTX)-blocked EPSCs were partially rescued by 4-aminopyridine (4-AP), indicating monosynaptic nature of connections in the pBLA to the vCA1^{Calb1+} and aBLA to the vCA1^{Calb1-} pathways. **(c, h)** Changes of EPSCs amplitude in TTX only and TTX + 4-AP. **(e, j)** Percentage of vCA1^{Calb1+} and vCA1^{Calb1-}

neurons in response to the photostimulation of pBLA–vCA1 (e) and aBLA–vCA1 pathways (j). n=13 vCA1^{Calb1+} neurons and 12 vCA1^{Calb1-} neurons from 10 pBLA–vCA1-ChR2 mice; n=11 vCA1^{Calb1+} and 11 vCA1^{Calb1-} neurons from 9 aBLA–vCA1-ChR2 mice.

4. A number of the papers they cite either broadly targeted BLA or vHPC and see the same optogenetic effects as they claim to for the aBLA projectors. Again given that multiple previous papers have claimed that aBLA is a far less significant population than the pBLA, it seems more likely that the previous results would have been primarily inhibiting the pBLA projectors, and that is what is suggested by the histology shown in some of those papers, particularly the Felix-Ortiz et al. 2013 paper. This should be discussed.

[Response]: Thanks to the reviewer for the comments. By carefully checking the coordinates in the papers of Tye et al., 2011¹ and Felix–Ortiz et al. 2013², we found that the injection sites “bregma: -1.6 mm anteroposterior, ±3.1 mm mediolateral, -4.5 mm dorsoventral” (Tye et al., 2011) and “bregma: -1.6 mm anteroposterior, ±3.2-3.4 mm mediolateral, -4.9 mm dorsoventral” (Felix–Ortiz et al. 2013) were precisely targeted aBLA not a/pBLA in mice. During photostimulation epoch, they found that aBLA and aBLA–vCA1 circuit exerted anxiogenic effect, which was also confirmed in our current study.

We totally agree with the reviewer that broad damage on BLA can inhibit pBLA and its projection to vCA1. However, the interpretation of final outcome should not be simply attributed to the relative strength between aBLA–vCA1 and pBLA–vCA1 connections, because BLA could orchestrate to modulate emotion *via* many downstreams. Previous papers have claimed that aBLA neurons send axons to vCA1², BNST³, mPFC⁴, CeA^{1, 5, 6}, Pir⁷ and Lent⁸. Among them, at least aBLA–innervated vCA1² and mPFC⁹ neurons had been proven to regulate anxiety–related behaviors, while aBLA–innervated BNST and CeA had been identified to exert anxiolytic effects^{1, 3} (please note that the above studies only used the term of BLA without further

identifying the subregions, we have identified the aBLA according to their injection site). Thus, the final outcome of the manipulation on a/pBLA population neurons should depend on the convergence effect of all the downstreams of a/pBLA. Now, we added this discussion in page 17 line 417 to 421.

5. The authors repeatedly generalize their findings in troubling ways, most of which might be due to their issues with the losing clarity in the writing, but needs to be properly addressed in the edits.

[Response]: Thanks to the reviewer for the comments. We have asked editing service (Springer Nature) to make further professional editing of our manuscript (highlighted in red). Now, we believe that the clarity of the paper has been substantially improved.

For example, “Although negative and positive emotion were respectively elicited after aBLA and pBLA manipulation...”

[Response]: We revised the sentence into “Manipulating anterior BLA (aBLA) and posterior BLA (pBLA or BLP) can elicit negative and positive emotional behaviours, respectively, but their heterogeneity at the molecular level is still unclear.” (please see page 16 line 394 to 396).

Again, there is nothing in the data that supports their claim that the two projections differ in their strength (their claim in figure 7).

[Response]: As suggested by the reviewer, we have deleted the expression of “a/pBLA–vCA1 circuit imbalance” in Figure 7 (please see Figure 7b).

[REDACTED]

Figure 7. Proposed working model.

(a) Heterogeneity in the BLA–vCA1 circuit under physiological conditions. The BLA shows proteomic diversity along its anterior–posterior axis at the molecular level. The anterior part of the BLA (aBLA) and the posterior BLA (pBLA) innervate the deep-layer calbindin1-negative neurons (Calb1⁻) and superficial-layer calbindin1-positive neurons (Calb1⁺) in vCA1, forming aBLA–vCA1 and pBLA–vCA1 circuits, respectively. This molecular and structural heterogeneity endows these pathways with functional heterogeneity in controlling approach–avoidance behaviour, *i.e.*, the aBLA–vCA1^{Calb1⁻} circuit promotes avoidance and exerts an anxiogenic effect, while the pBLA–vCA1^{Calb1⁺} circuit triggers approach and exerts an anxiolytic effect. (b) In AD, different protein network changes in response to A β deposition in the aBLA and pBLA impair the aBLA–vCA1^{Calb1⁻} and pBLA–vCA1^{Calb1⁺} circuits. Together with their disorganized firing patterns, the AD mice prefer avoidance over approach in conflict tasks and display anxiety. Furthermore, Calb1 expression determines the anxiolytic effect of pBLA–vCA1^{Calb1⁺} stimulation in AD, indicating a molecular mechanism at the exit node of the pBLA–vCA1^{Calb1⁺} circuit.

Reference #1 seems like a strange choice when making a broad claim about emotion and decision making.

[Response]: We deeply appreciate the reviewer's comment and apologize for the inappropriate citations. Now, we have deleted the Reference #1 in the revised version.

6. This was brought up in the first round, It's not clear that the second half of the paper adds much to the narrative and is a bit hard to interpret. To make sense of the shRNA manipulation before doing this test in the APP/PS1 mice they should've done a control in their WT cohorts to see if the shRNA injection has the same effect on the optogenetic anxiogenic and anxiolytic effects.

[Response]: We thank the reviewer for the suggestions. In the first round of review, we have supplemented the data of shRNA in Wt cohorts as suggested. Please see sFigure 16 and page 5 line 101, page 14 line 337 to 346 and page 19 line 475. We have also deleted the part of the narrative in the second half of the paper to avoid unnecessary verbosity and confusion. Please see page 14 line 337 and 378.

7. SFig13 doesn't seem to add much to the study. Of course stimulation is going to increase the number of cells expressing Fos, not clear what this means for the APP/PS1 phenotype.

[Response]: We appreciate for the opportunity to clarify this point. By c-Fos staining, we observed that the number of Calb1^{+/c}-Fos⁺ neurons in the vCA1 was significantly decreased in the APP/PS1 mice compared with the levels in the age- and sex-matched wild-type controls (sFigure 15a,b). These data indicate insufficient activation of Calb1⁺ neurons in AD mice. Considering the monosynaptic connection between pBLA and vCA1^{Calb1⁺} (Figure 1, sFigure 6) and the inhibition of pBLA-vCA1 pathway in APP/PS1 mice (Figure 6), we then photostimulated pBLA-vCA1 inputs to identify whether targeting pBLA-vCA1 connection could rescue the insufficient activation of vCA1^{Calb1⁺} in AD mice. Not surprisingly, the number of Calb1^{+/c}-Fos⁺ neurons in the vCA1 of the APP/PS1 mice was robustly increased after pBLA-vCA1 input targeting (sFigure 15a,b). Together with the anxiolytic effect of vCA1^{Calb1⁺} in wild-type cohorts (sFigure 12) and the anti-anxiety effect of pBLA-vCA1 inputs in AD mice (Figure 6), we concluded that activation of vCA1^{Calb1⁺} is the mechanism of anti-anxiety of pBLA-

vCA1 input manipulation in AD. Thus, we hope the reviewer agree that these expected results are also meaningful, because those data and the results from vCA1-Calb1-ChR2 mice and pBLA-vCA1-ChR2 mice could complement each other and explain the pathogenesis of anxiety in AD more completely.

Reviewer #2 (Remarks to the Author):

The authors have addressed my comments and suggestions for improvement.

We sincerely appreciate the reviewer's positive comments on our manuscript.

Reviewer #3 (Remarks to the Author):

Overall the authors have done a good job at addressing the reviewer's concerns. Only a few problems remain unsolved. Despite these small problems, the results in general are interesting and I believe the manuscript can be accepted as it is.

[Response]: We greatly appreciate the reviewer for the positive evaluation of our work.

1. The writing is better, but the quality is still considerably below average for this journal. In the initial submission the writing quality sometimes precluded understanding of what the authors wanted to communicate. Now it is possible to understand everything, but there are still many instances of awkward and incorrect phrasing (such as “disorderly firing”). For example, the labels in Fig 5 are very strange. What is “single organism process”? Fig 7a should use “Structural heterogeneity” and “functional heterogeneity” instead of “structure” and “function heterogeneity”.

[Response]: We thank the reviewer for the constructive comments. We have asked editing service (Springer Nature) to make further professional editing of our manuscript (highlighted in red). Now, we believe that the clarity of the paper has been substantially improved.

In the revised version, we have replaced the expression of “disorderly firing” with “disorganized firing” (please see Figure 7). Also, we have used “Structural heterogeneity” and “Functional heterogeneity” instead of “structure” and “function heterogeneity” as the reviewer suggested (please see Figure 7).

[REDACTED]

Figure 7. Proposed working model.

(a) Heterogeneity in the BLA–vCA1 circuit under physiological conditions. The BLA shows proteomic diversity along its anterior–posterior axis at the molecular level. The anterior part of the BLA (aBLA) and the posterior BLA (pBLA) innervate the deep-layer calbindin1-negative neurons (Calb1⁻) and superficial-layer calbindin1-positive neurons (Calb1⁺) in vCA1, forming aBLA–vCA1 and pBLA–vCA1 circuits, respectively. This molecular and structural heterogeneity endows these pathways with functional heterogeneity in controlling approach–avoidance behaviour, *i.e.*, the aBLA–vCA1^{Calb1⁻} circuit promotes avoidance and exerts an anxiogenic effect, while the pBLA–vCA1^{Calb1⁺} circuit triggers approach and exerts an anxiolytic effect. (b) In AD, different protein network changes in response to A β deposition in the aBLA and pBLA impair the aBLA–vCA1^{Calb1⁻} and pBLA–vCA1^{Calb1⁺} circuits. Together with their disorganized firing patterns, the AD mice prefer avoidance over approach in conflict tasks and display anxiety. Furthermore, Calb1 expression determines the anxiolytic effect of pBLA–vCA1^{Calb1⁺} stimulation in AD, indicating a molecular mechanism at the exit node of the pBLA–vCA1^{Calb1⁺} circuit.

The term of “single organism process” (GO:0044699) comes from the database of Gene Ontology. We feel very sorry that we have not found the original basis for the definition of “single organism process”. However, in GRAMENE website (<https://archive.gramene.org/db/ontology/search?id=480667>), it is defined as a biological process that involves only one organism. The subcatalog of “single organism process” includes antibody-dependent cellular cytotoxicity (GO:0001788), T cell mediated cytotoxicity (GO:0001913), sulfur utilization (GO:0006791), *etc.*. In our study, the differential proteins in “single organism process” included Abi2, Hebp1, Camk2n1, *etc.*. By KEGG analysis, we found that they were enriched in the cellular metabolic process, the response to stimulus, regulation of protein phosphorylation, *etc.*.

2. Fig 1i and 1j. Please add data showing the responses are blocked after adding TTX (TTX alone, in the absence of 4-AP).

[Response]: We thank the reviewer for pointing out this issue. As suggested, we have added TTX without 4-aminopyridine (4-AP) into the ACSF and found that the evoked EPSPs and EPSCs in pBLA-vCA1^{Calb1+} pathway were completely blocked (Figure 1i, j and sFigure 6b, c, g, h). These data indicate that ChR2-mediated depolarization by photostimulation alone is insufficient to induce neurotransmitter release from nerve terminal. However, the TTX-blocked EPSPs and EPSCs were partially rescued in the presence of 4-AP (Figure 1i, j and sFigure 6b, c, g, h). These are due to the inhibitory effect of 4-AP on voltage-gated K⁺ channels, through which additional depolarization may be induced at the terminal, so that ChR2-mediated depolarization may now be sufficient to trigger glutamate release even in the presence of TTX¹⁰, leading to the generation of responses in postsynaptic vCA1^{Calb1+} neurons. Since the rescue of TTX-blocked EPSCs by 4-AP may be observed only if ChR2-expressing fibers project monosynaptically to postsynaptic neurons¹¹, our new data from TTX and TTX+4AP strongly suggest a monosynaptic nature of pBLA-vCA1^{Calb1+} pathway. We also extended the results part of this supplementary data on page 7, lines 144–157.

Figure 1. Features of aBLA- and pBLA-innervated vCA1 neurons along the superficial–deep axis.

(a, b) Representative images (a) and quantification (b) show the predominance of aBLA-innervated neurons in the deep layer of vCA1 (left) and pBLA-innervated neurons in the superficial layer (right). By anterograde monosynaptic tracing, vCA1 neurons receiving monosynaptic inputs from aBLA and pBLA were labelled by tdTomato. Scale bar, 100 μ m; n=4 per group. **(c, d)** Representative images (c) and quantification (d) show simple (c, left) and complex (c, right) vCA1 neurons innervated by aBLA and pBLA. Scale bar, 30 μ m; n=5 per group. **(e–g)** Distinct distributions of calbindin1-positive neurons (Calb1+) in aBLA- and pBLA-innervated vCA1 neurons. By anterograde multisynaptic tracing (H129–G4), pBLA-innervated vCA1 neurons (GFP) were predominately colocalized with Calb1+(tdTomato) (f, g), while low colocalization was shown in aBLA-innervated vCA1 neurons (e, g). Scale bar, 100 μ m; n=7 per group. Unpaired *t*-test, $t=24.76$ $df=7.266$, $P<0.0001$. **(h)** Action potential firing

of a pBLA neuron in response to patterned blue laser light (473 nm, 5/10/20/50 Hz, 5 ms pulses) recorded by *ex vivo* current-clamp recording. Scale bar, 20 mv and 500 ms. (i, j) Subthreshold responses of vCA1 neurons to the photoactivation of pBLA-vCA1 inputs, 20 Hz, 5 ms pulses in ACSF (top) with TTX, TTX+4AP or TTX+4AP+AP5+NBQX (GluR antagonist). Scale bar, 2 mv and 200 ms (i). Amplitude changes in EPSPs after TTX, TTX+4AP or GluR antagonist perfusion (one-way ANOVA, $F(3, 48) = 155.1$, $P < 0.0001$, Tukey's post hoc analysis, $P < 0.01$) (j). $n=13$ cells from 7 mice. Data are presented as the mean \pm SEM.

Supplementary Figure 6. Identification of monosynaptic pBLA-vCA1^{Calb1+} and aBLA-vCA1^{Calb1-} connections. In Calb1-IRES2-Cre-D::Ai9 mice, AAV5-CaMKIIa-hChR2(H134R)-EYFP was injected into the aBLA and pBLA respectively. Four weeks later, electrophysiological response was recorded from vCA1^{Calb1+} and vCA1^{Calb1-} neurons upon the photoactivation of pBLA-vCA1 inputs or aBLA-vCA1 inputs. (a, f) Representative images of patch pipette tips on vCA1^{Calb1+} (tdTomato+) and vCA1^{Calb1-} (tdTomato-) pyramidal neurons in hippocampal slices (scale bar=30 μ m). (b, d, g, i)

Representative traces of EPSCs in the pBLA–vCA1^{Calb1+} pathway (**b, d**) and aBLA–vCA1^{Calb1-} pathway (**g, i**) recorded under different experimental conditions. EPSCs were evoked by photostimulation of ChR2-expressing axons from the pBLA and aBLA, and recorded in Calb1⁺ and Calb1⁻ neurons respectively in the vCA1. Optogenetically-induced and tetrodotoxin (TTX)-blocked EPSCs were partially rescued by 4-aminopyridine (4-AP), indicating monosynaptic nature of connections in the pBLA to the vCA1^{Calb1+} and aBLA to the vCA1^{Calb1-} pathways. (**c, h**) Changes of EPSCs amplitude in TTX only and TTX + 4-AP. (**e, j**) Percentage of vCA1^{Calb1+} and vCA1^{Calb1-} neurons in response to the photostimulation of pBLA–vCA1 (**e**) and aBLA–vCA1 pathways (**j**). n=13 vCA1^{Calb1+} neurons and 12 vCA1^{Calb1-} neurons from 10 pBLA–vCA1-ChR2 mice; n=11 vCA1^{Calb1+} and 11 vCA1^{Calb1-} neurons from 9 aBLA–vCA1-ChR2 mice.

References

1. Tye KM, *et al.* Amygdala circuitry mediating reversible and bidirectional control of anxiety. *Nature* **471**, 358-362 (2011).
2. Felix-Ortiz AC, Beyeler A, Seo C, Leppla CA, Wildes CP, Tye KM. BLA to vHPC inputs modulate anxiety-related behaviors. *Neuron* **79**, 658-664 (2013).
3. Kim SY, *et al.* Diverging neural pathways assemble a behavioural state from separable features in anxiety. *Nature* **496**, 219-223 (2013).
4. Padilla-Coreano N, *et al.* Direct Ventral Hippocampal-Prefrontal Input Is Required for Anxiety-Related Neural Activity and Behavior. *Neuron* **89**, 857-866 (2016).
5. Kim J, Zhang X, Muralidhar S, LeBlanc SA, Tonegawa S. Basolateral to Central Amygdala Neural Circuits for Appetitive Behaviors. *Neuron* **93**, 1464-1479 e1465 (2017).
6. Beyeler A, *et al.* Divergent Routing of Positive and Negative Information from the Amygdala during Memory Retrieval. *Neuron* **90**, 348-361 (2016).
7. Sadrian B, Wilson DA. Optogenetic Stimulation of Lateral Amygdala Input to Posterior Piriform Cortex Modulates Single-Unit and Ensemble Odor Processing. *Frontiers in neural circuits* **9**, 81 (2015).
8. McDonald AJ, Zaric V. GABAergic somatostatin-immunoreactive neurons in the amygdala project to the entorhinal cortex. *Neuroscience* **290**, 227-242 (2015).
9. Felix-Ortiz AC, Burgos-Robles A, Bhagat ND, Leppla CA, Tye KM. Bidirectional modulation of anxiety-related and social behaviors by amygdala projections to the medial prefrontal cortex. *Neuroscience* **321**, 197-209 (2016).
10. Petreanu L, Mao T, Sternson SM, Svoboda K. The subcellular organization of neocortical excitatory connections. *Nature* **457**, 1142-1145 (2009).
11. Cho JH, Deisseroth K, Bolshakov VY. Synaptic encoding of fear extinction in mPFC-amygdala circuits. *Neuron* **80**, 1491-1507 (2013).

REVIEWERS' COMMENTS:

Reviewer #1 (Remarks to the Author):

By adding a quantification of viral spread across AP axis and whole cell recording of the Calb+/- cells in vHPC the authors have addressed my concerns. This is an exciting paper, that further increases our understanding of the vHPC circuit in driving anxiety related behavior.

Reviewer #3 (Remarks to the Author):

The authors have adequately addressed all my concerns.

REVIEWERS' COMMENTS:

Reviewer #1 (Remarks to the Author):

By adding a quantification of viral spread across AP axis and whole cell recording of the Calb+/- cells in vHPC the authors have addressed my concerns. This is an exciting paper, that further increases our understanding of the vHPC circuit in driving anxiety related behavior.

[Response]: We greatly appreciate the reviewer for the positive evaluation of the manuscript.

Reviewer #3 (Remarks to the Author):

The authors have adequately addressed all my concerns.

[Response]: We thank the reviewer very much for the positive comments of our present work.